# Structural basis for the preservation of a subset of topologically associating domains in interphase chromosomes upon cohesin depletion

Davin Jeong[1], Guang Shi[1], Xin Li[1], D Thirumalai[1,2]*

[1]Department of Chemistry, University of Texas at Austin, Austin, United States; [2]Department of Physics, University of Texas at Austin, Austin, United States

**Abstract** Compartment formation in interphase chromosomes is a result of spatial segregation between euchromatin and heterochromatin on a few megabase pairs (Mbp) scale. On the sub-Mbp scales, topologically associating domains (TADs) appear as interacting domains along the diagonal in the ensemble averaged Hi-C contact map. Hi-C experiments showed that most of the TADs vanish upon deleting cohesin, while the compartment structure is maintained, and perhaps even enhanced. However, closer inspection of the data reveals that a non-negligible fraction of TADs is preserved (P-TADs) after cohesin loss. Imaging experiments show that, at the single-cell level, TAD-like structures are present *even without cohesin*. To provide a structural basis for these findings, we first used polymer simulations to show that certain TADs with epigenetic switches across their boundaries survive after depletion of loops. More importantly, the three-dimensional structures show that many of the P-TADs have sharp physical boundaries. Informed by the simulations, we analyzed the Hi-C maps (with and without cohesin) in mouse liver and human colorectal carcinoma cell lines, which affirmed that epigenetic switches and physical boundaries (calculated using the predicted 3D structures using the data-driven HIPPS method that uses Hi-C as the input) explain the origin of the P-TADs. Single-cell structures display TAD-like features in the absence of cohesin that are remarkably similar to the findings in imaging experiments. Some P-TADs, with physical boundaries, are relevant to the retention of enhancer–promoter/promoter–promoter interactions. Overall, our study shows that preservation of a subset of TADs upon removing cohesin is a robust phenomenon that is valid across multiple cell lines.

*For correspondence:
dave.thirumalai@gmail.com

Competing interest: The authors declare that no competing interests exist.

## eLife assessment

This **valuable** study, of interest for students of the biology of genomes, uses simulations in combination with published data to examine how many TADs remain after cohesin depletion. The authors suggest that a significant subset of chromosome conformations do not require cohesin, and that knowledge of specific epigenetic states can be used to identify regions of the genome that still interact in the absence of cohesin. The theoretical approaches and quantitative analysis are state-of-the-art, and the data quality and strength of the conclusions are **convincing**, but it is unfortunately still unclear whether physical boundaries (of domains?) in the model appear to be a consequence of preserved TADs, or whether preserved TADs are caused by the physical boundaries.

## Introduction

Advances in experimental techniques have provided glimpses of the three-dimensional (3D) organization of chromosomes in diverse species (*Fraser et al., 2015b*; *Wang et al., 2016*; *Boettiger et al., 2016*; *Rao et al., 2014*; *Dekker et al., 2013*; *Lieberman-Aiden et al., 2009*). The average (performed over a large number of cells) contact map (*Rao et al., 2014*; *Lieberman-Aiden et al., 2009*), inferred using chromosome conformation capture technique and related variants (hereon referred to as Hi-C), is a two-dimensional (2D) matrix, whose elements are a measure of the probability that two loci separated by a certain genomic distance are spatially adjacent. The Hi-C experiments on different mammalian cells suggest that there are two major length scales in the organization of interphase chromosomes. On the scale, $L_C \sim$ (2-5) Mbp (megabase pairs), one observes checkerboard patterns in the contact maps (*Rao et al., 2014*), which are thought to be associated with micro-phase separation between the two major epigenetic states, active (A) or euchromatin and inactive (B) or heterochromatin. On the length scale, $L_{TAD}$, from tens of kb up to a few Mb, domains, referred to as topologically associating domains (TADs), appear as squares along the diagonal of the contact maps (*Dixon et al., 2012*; *Szabo et al., 2019*). Contacts are enriched within the TADs and are suppressed across the boundaries between the TADs. A number of polymer models (*Barbieri et al., 2012*; *Di Pierro et al., 2016*; *Shi et al., 2018*; *Conte et al., 2020*; *Jost et al., 2014*; *Barbieri et al., 2017*) have shown that compartment formations and TADs may be explained using micro-phase separation between A- and B-type loci. The use of two length scales, $L_C$ and $L_{TAD}$, in characterizing the organization of interphase chromosomes is now entrenched in the field, although there are suggestions that finer sub-TAD structures emerge at kilobase scales (*Phillips-Cremins et al., 2013*; *Fraser et al., 2015a*). In particular, recent Micro-C experiments have shown that there are fine structures starting from the nucleosome level (*Hsieh et al., 2015*; *Hsieh et al., 2016*; *Hsieh et al., 2020*), thus establishing the hierarchical organization of interphase chromosomes over a broad range of length scales.

TADs are thought to regulate gene expression by constraining the contacts between target gene and regulatory regions (*Dowen et al., 2014*; *Özdemir and Gambetta, 2019*). As a consequence, perturbation or disruption of their integrity such as deletions, duplications, or inversions of DNA segments within the TADs could lead to aberrant gene expression (*Hnisz et al., 2016a*; *Szabo et al., 2019*; *Ji et al., 2016*; *Lupiáñez et al., 2015*; *Hnisz et al., 2016b*; *Rao et al., 2017*; *Ren and Dixon, 2015*; *Bianco et al., 2018*). A class of chromatin loops, mediated by the ATP-dependent motor cohesin (*Kim and Yu, 2020*) and the DNA-binding protein CTCF protein ('cohesin-associated CTCF loop'), organizes a subset of the TADs (*Fudenberg et al., 2016*). It is thought that cohesin (*Kim and Yu, 2020*) extrudes DNA loops of varying lengths, which are terminated when the motor encounters the transcriptional insulator CCCTC-binding factor (CTCF) (*Phillips and Corces, 2009*). This implies that cohesin and CTCF are often colocalized at the TAD boundary (*Fudenberg et al., 2016*; *Lieberman-Aiden et al., 2009*; *Guo et al., 2015*; *Rao et al., 2014*; *Dekker et al., 2013*; *Vietri Rudan et al., 2015*).

Several experiments have shown that depletion of the architectural proteins (Nipbl, RAD21, and CTCF) disrupts the organization of interphase chromosomes (*Rao et al., 2017*; *Schwarzer et al., 2017*; *Nuebler et al., 2018*; *Wutz et al., 2017*; *Haarhuis et al., 2017*; *Nora et al., 2017*; *Zuin et al., 2014*; *Bintu et al., 2018*; *de Wit and Nora, 2023*). *Schwarzer et al., 2017* showed that the removal of the cohesin loading factor, *Nipbl*, in the mouse liver cell results in loss of TADs. They concluded that compartment formation, which is independent of cohesin, is a consequence of the underlying epigenetic landscape, while TAD formation requires cohesin. Similarly, it was found that upon removal of cohesin subunit *RAD21* cohesin-associated CTCF loops and TADs are abolished (*Rao et al., 2017*; *Zuin et al., 2014*; *Bintu et al., 2018*). Deletion of *RAD21* results in the complete loss of the so-called loop domains (*Rao et al., 2017*), which are formed when CTCF colocalizes with cohesin. In contrast, imaging experiments (*Bintu et al., 2018*) showed that TAD-like structures, with sharp boundaries, at the single-cell level survive even after deleting cohesin. Three points emerged from these studies. (1) They reinforce the two-length-scale description of genome organization at the ensemble level. (2) Factors that prevent the association of cohesin with chromosomes globally abolish the TADs and the Hi-C peaks, but preserve (or even enhance) compartmentalization. Experimental studies (*Schwarzer et al., 2017*; *Rao et al., 2014*; *Zuin et al., 2014*; *Bintu et al., 2018*; *Rao et al., 2017*) and polymer simulations (*Nuebler et al., 2018*; *Conte et al., 2020*; *Xie et al., 2020*) have shown that the global epigenetic state determines compartment formation, while the more dynamic

TADs, with finite lifetimes (*Hansen et al., 2017*), require ATP-dependent cohesin. (3) TAD-like features persist in single cells before and after auxin treatment, albeit with changes in the locations of the sharp domain boundaries.

The results of super-resolution experiments (*Bintu et al., 2018*) at the single-cell level (described above) made us wonder whether there is evidence for the preservation of TADs at the ensemble level upon cohesin depletion. To this end, we first analyzed the experimental contact maps from mouse liver and HCT-116 cells (human colorectal carcinoma cell line) in the presence and absence of cohesin to assess whether TADs are preserved. We discovered that, on average, a fraction of TADs, identified using the TopDom method (*Shin et al., 2016*), are retained in chromosomes from both the cell lines (*Figure 1*) after removing cohesin. These findings raise the following questions. What is the mechanistic basis for the retention of a small but significant fraction of TADs that are preserved after cohesin loss? Is there a structural explanation for TAD retention at the ensemble level (Hi-C), which would reconcile with the results in super-resolution imaging experiments showing TAD-like structures at the single level, even without cohesin?

We answered the questions posed above by using the following strategy. We first performed polymer simulations for two chromosomes from the GM12878 cell line using the chromosome copolymer model (CCM) (*Shi et al., 2018*) with and without loop anchors, which mimics the wild-type (WT) and the absence of cohesin-associated CTCF loops, respectively. The major purpose of the CCM polymer simulations is to determine the mechanisms for the emergence of preserved TADs (P-TADs). Because the simulations directly generate 3D structures, they can be used to compute average contact maps that can be compared with experiments to determine the accuracy of the CCM. In addition, comparisons of the contact maps with and without cohesin allowed us to generate the mechanisms for the emergence of P-TADs. Using the polymer simulations of chromosomes from the GM12878 cell line (*Rao et al., 2014*), whose organization without cohesin is unknown, we determined that P-TADs arise due to epigenetic switches across TAD boundaries and/or associated with peaks in boundary probabilities, which require knowledge of ensemble of 3D structures.

Informed by the results from the polymer simulations, we analyzed the experimental data from two cell lines (*Schwarzer et al., 2017*; *Rao et al., 2017*). We discovered that epigenetic switch does account for a reasonable fraction ($\approx 0.4$ in mouse liver and $\approx 0.3$ in HCT-116 cell lines) of P-TADs. Rather than perform multiple time-consuming polymer simulations, we generated the 3D structural ensemble of chromosomes using the accurate and data-driven Hi-C-polymer-physics-structures (HIPPS) method (*Shi and Thirumalai, 2021*), utilizing the experimental Hi-C data. The analyses using the 3D structures accounted for about 53% of the P-TADs, predicted by the TopDom method (*Shin et al., 2016*). Strikingly, the 3D structures revealed TAD-like structures at the single-cell level both in the presence and absence of cohesin, which is in accord with the super-resolution imaging data (*Bintu et al., 2018*). Our work shows that the effects of cohesin removal on chromatin structures are nuanced, requiring analyses of both the epigenetic landscape and 3D structures in order to obtain a comprehensive picture of how distinct factors determine interphase chromosome organization in the nucleus. Our calculations for chromosomes from three cell lines lead to the robust conclusion that a subset of P-TADs is intact after depletion of cohesin.

## Results

### A non-negligible fraction of TADs is preserved upon removal of cohesin

Experiments (*Schwarzer et al., 2017*; *Rao et al., 2017*; *Wutz et al., 2017*; *Zuin et al., 2014*; *Bintu et al., 2018*; *Nuebler et al., 2018*; *Haarhuis et al., 2017*) have shown that deletion of cohesin loaders (*Nipbl in mouse liver, SCC2 in yeast*) and cohesin subunit (*RAD21*) abolishes a substantial fraction of both cohesin-associated CTCF loops and TADs. These observations across different cell lines raise an important question: Do all the TADs completely lose their contact patterns after removal of cohesin? To answer this question, we first analyzed 50kb-resolution contact maps from the two cell lines (mouse liver *Schwarzer et al., 2017* and HCT-116 *Rao et al., 2017*) before and after degradation of *Nipbl* and *RAD21*, respectively (see section 'Analyses of the experimental data' for details). Using TopDom (*Shin et al., 2016*), we discovered that roughly 659 TADs out of 4176 (16%) are preserved (*Figure 1a*) after removing *Nipbl* in the mouse liver cells. In the HCT-116 cells, 1226 TADs out of 4733 (26%) are preserved (*Figure 1b*) upon *RAD21* loss. *Figure 1c and d* show that the number of P-TADs depends

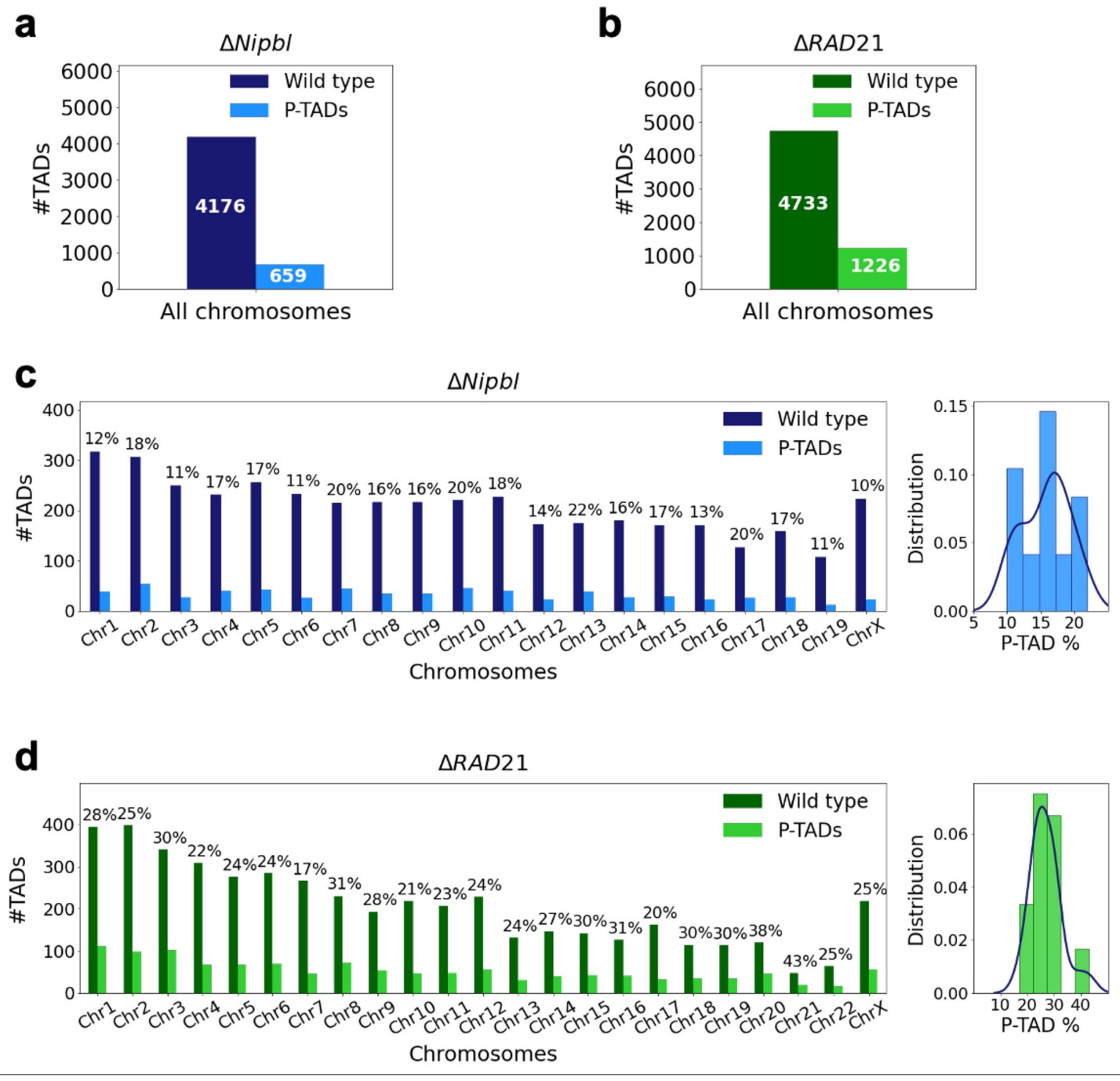

**Figure 1.** Fate of the topologically associating domains (TADs) in chromosomes upon cohesin deletion. (**a**) The number of TADs in all the chromosomes, identified by the TopDom method (Shin et al. 2015), in the wild-type (WT) cells and the number of preserved TADs (P-TADs) after deleting cohesin loading factor (Δ*Nipbl*) in mouse liver. (**b**) Same as (**a**) except the experimental data are analyzed for HCT-116 cell before (WT) and after RAD21 deletion. (**c**) The total number of TADs and the number of P-TADs for each chromosome calculated using the mouse liver Hi-C data. The number above each bar is the percentage of P-TADs in each chromosome. (**d**) Same as (**c**) except the results are for chromosomes from the HCT-116 cell line. The percentage of P-TADs is greater in the HCT-116 cell line than in mouse liver for almost all the chromosomes, a feature that is more prominent in the distribution of P-TAD proportions (right).

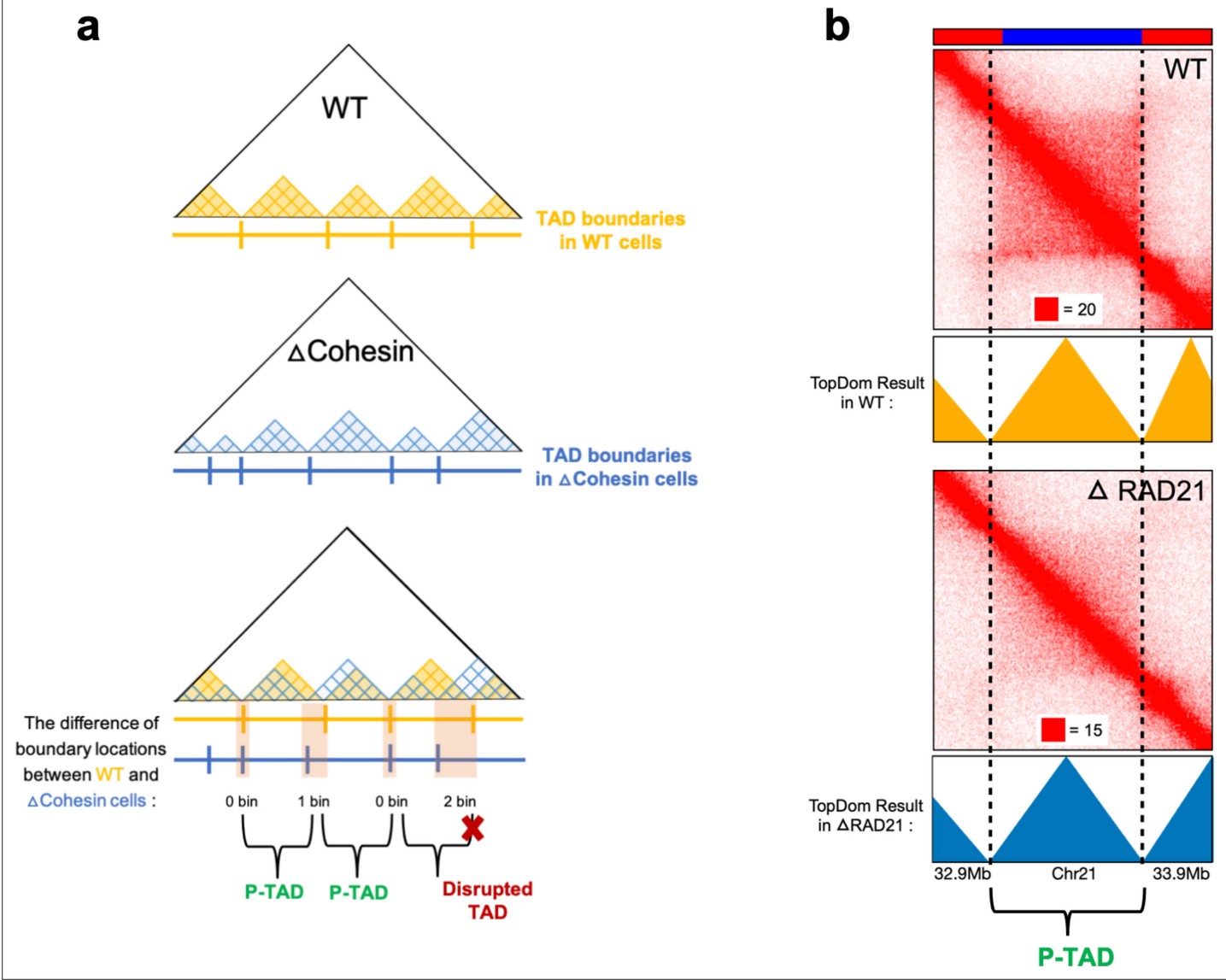

**Figure 2.** Identification of preserved topologically associating domains (P-TADs) from the contact map using the TopDom method. (**a**) Schematic representation used to determine the P-TADs. Yellow (blue) triangles represent the TADs identified using the TopDom method in wild-type (WT) (cohesin-depleted) contact maps at 50 kb resolution. Small square within each triangle represents a single locus (50 kb size). The boundaries of a TAD detected in the WT contact map within ± one bin (50 kb) from a position of boundaries in cohesin-depleted cells are deemed to be a P-TAD. (**b**) P-TAD upon cohesin loss in HCT-116 cell. The bar plots above the contact maps show the epigenetic states. Red (blue) color represents the active (inactive) state. The TAD between gray dashed lines is preserved upon cohesin loss. The parameter (with red square) displayed at each left bottom indicates the color scale when plotting contact maps used in Juicebox (*Robinson et al., 2018*).

on the chromosome number. Although the actual number of P-TADs would depend on the TAD-calling protocol (see *Figure 2* and section 'TAD and P-TAD identification'), the finding that a non-negligible fraction is preserved after cohesin depletion is highly significant.

## CCM simulations reproduce wild-type Hi-C maps

To explore the mechanism resulting in P-TADs, we first simulated the CCM (*Shi et al., 2018*; *Appendix 5—figure 2a*). To independently decipher the origins of P-TADs (*Figure 1*) in experiments, we calculated the contact maps for Chr13, shown in *Appendix 5—figure 2* (Chr10 in *Appendix 5—figure 3*) from the GM12878 cell line. The CCM simulations (*Appendix 5—figure 2b*) reproduce the ubiquitous checkerboard patterns well. The rectangle in *Appendix 5—figure 2b* represents the border of one such compartment formed primarily by the interactions between the B-type loci.

In order to quantitatively compare the Hi-C data and the simulated contact maps, we transformed the contact maps into Pearson correlation maps, which are used to assess whether two loci have correlated interaction profiles (*Appendix 5—figure 2c*). The Kullback–Leibler (KL) divergence between the two probability distributions for the Pearson correlation coefficients (PCCs), $\rho_{ij}$ s, from simulations and experiments is 0.04 (see *Appendix 5—figure 2d*). We also performed principal component analysis (PCA) on the Pearson correlation matrix to identify the compartment structure. A comparison of the PCA-derived first principal components (PC1) across the Chr13 reveals that A/B compartments observed in the CCM correspond well to those found in the experiments (*Appendix 5—figure 2e*).

We then compared the 3D spatial organization between the simulations and experiments using the Ward linkage matrix (WLM), which is based on an agglomerative clustering algorithm. The simulated WLM is calculated from a spatial distance map of the organized chromosome (described in *Appendix 5—figure 2f–h*). We constructed the experimental WLM by converting the Hi-C contact map into a distance map using the approximate relationship (*Wang et al., 2016*; *Shi et al., 2018*; *Shi and Thirumalai, 2021*), $P_{ij} \propto R_{ij}^{-4.1}$. Here, $P_{ij}$ is the contact probability, and $R_{ij}$ is the mean 3D spatial distance between loci $i$ and $j$ (see section 'Ward linkage matrix'). The PCC between experimental and simulated WLMs is 0.83 (*Appendix 5—figure 2h*), which establishes the accuracy of the CCM.

Snapshots of TAD structures in *Appendix 5—figure 2k and n* show that they are compact but structurally diverse. The average length of the TADs detected using TopDom (*Shin et al., 2016*) from Hi-C and simulated contact maps are ~615 kbs and ~565 kbs, respectively. Overall the emerging picture of the compartment and TAD structures using different methods is consistent with each other. The results in *Appendix 5—figure 2* show that the agreement between the CCM simulations and Hi-C data is excellent, especially considering that (1) error estimates in the Hi-C experiments are essentially unknown, and (2) more importantly, only a single parameter, the inter-loci interaction strength, $\epsilon$, is tuned to fit the experimental contact maps (see Methods sections). Taken together, the results show that the key features of the Hi-C maps for Chr13 (see *Appendix 5—figure 3* for Chr10 results) are accurately reproduced by the CCM simulations.

## Epigenetic switch accounts for a large fraction of P-TADs

Most of the TADs are not discernible after loop loss, as evidenced by the blurred edges in the contact maps (*Figure 3c*). In the CCM simulations of chromosomes from the human GM12878 cell line, a subset of TADs remains even after deleting cohesin-associated CTCF loops (*Figure 3a*). The percentages of P-TADs depend on both the resolution of the Hi-C experiments and the algorithm used to identify the TADs. By using the same method to analyze both the simulation results and experimental data, it is hoped that the conclusions would be robust.

We used the simulation results to determine the mechanism for the emergence of P-TADs by comparing the results for $P_L = 1$ and $P_L = 0$. The first observation is that some TADs, even with cohesin-associated CTCF loops, consist mostly of sequences in the same epigenetic state (A or B). *Figure 3d* compares the fate of one such TAD in the region (19.3–21.8 Mb) in Chr13 between $P_L = 1$ and $P_L = 0$. The highlighted TAD is preserved upon loop loss, although the probabilities of contact within this TAD are reduced when $P_L = 0$ (bottom) compared to $P_L = 1$ (top). Their boundaries correspond to a switch in the epigenetic state (the sequence location where the change in the epigenetic states occurs, as shown by the two black arrows). In contrast, a TAD in *Figure 3e*, which is present in the WT, is abolished when $P_L = 0$. The disruption of this particular TAD, lacking in at least one epigenetic switch, occurs because it can interact more frequently with neighboring TADs composed of similar epigenetic states, which in this case is B-type loci. The importance of epigenetic switch in the P-TADs has been noted before (*Rao et al., 2017*).

The results in *Figure 3d and e* show that a switch in the epigenetic state across a TAD boundary in the WT is likely to result in its preservation after cohesin-associated CTCF loop loss. To test this assertion, we calculated the number of P-TADs that are associated with switches in the epigenetic states (see section 'Data analyses' and *Appendix 5—figure 1* for details). We considered the TAD boundary and epigenetic switch as overlapping if they are less than 100kb apart, which is reasonable given that the Hi-C resolution adopted here is 50kb. By using 100kb as the cutoff, we only consider switch occurrences that exceed two loci. With this criterion, out of 216 (169) TADs calculated using TopDom, 50 (23) are P-TADs for Chr10 (Chr13) (vertical blue bars in *Figure 3b* in which there are epigenetic switches in the WT).

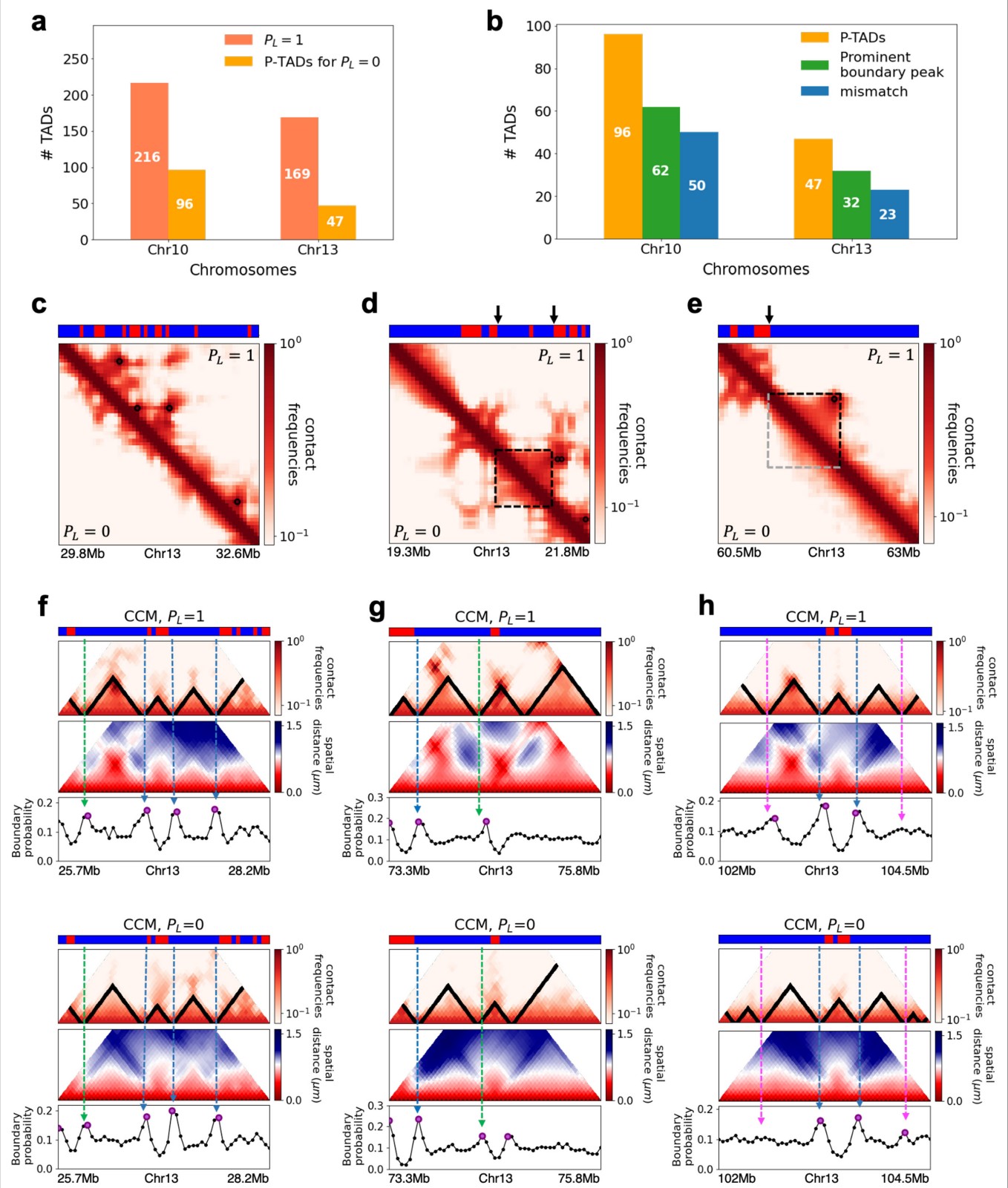

**Figure 3.** Chromosome copolymer model (CCM) simulations reveal characteristics of preserved topologically associating domains (P-TADs). (**a**) The number of TADs in the simulated Chr10 and Chr13 chromosomes for $P_L = 1$. The number of P-TADs after CTCF loop depletion ($P_L = 0$) is also shown. (**b**) The number of P-TAD with epigenetic switches (blue) and those identified by the peaks in the boundary probability (green). (**c–e**) Comparison between contact maps for the region of Chr13 with upper (lower) triangle with $P_L = 1$ ($P_L = 0$). The black circles at the corner of the TADs are the

*Figure 3 continued on next page*

*Figure 3 continued*

CTCF loop anchors. The bars above the contact map are the epigenetic states with red (blue) representing A (B) loci. Arrows above the bar show the epigenetic switch. (**c**) After loop deletion, TAD structures disappear. (**d**) TAD whose boundaries are marked by epigenetic switches are preserved. (**e**) TAD lacking at least one epigenetic switch is disrupted after loop loss. (**f–h**) Comparison of the contact map and the mean spatial distance matrices for the 2.5 Mb genomic regions (25.7–28.2 Mbp, 73.3-75.8 Mbp, and 102–104.5 Mbp, respectively) with (upper) and without (lower) loop anchors. Bottom graph shows the boundary probability, with the high values indicating population averaged TAD boundary. Purple circles in the boundary probability graph represent the preferred boundaries. A subset of P-TADs boundaries matches epigenetic switches (blue lines). P-TADs with high boundary probability is shown by the green line. The magenta line describes P-TADs, which are not accounted for by epigenetic switch or physical boundary in 3D space but are found using the TopDom method.

The P-TADs with epigenetic switches, illustrated in *Figure 3f*, show TADs in the 2.5 Mbs region in Chr13. Among the three P-TADs, two, whose boundaries are marked by dashed blue lines, have an epigenetic switch across the TAD boundary. These two TADs survive after the removal of cohesin-associated CTCF loop.

## P-TADs have prominent spatial domain boundaries

Because there are a number of P-TADs that are preserved *even without epigenetic switches* across their boundaries, we wondered whether the distance matrix, which requires 3D structures, would offer additional insights about P-TADs upon cohesin-associated CTCF loop loss. Recent imaging experiments (*Bintu et al., 2018*; *Cheng et al., 2021*; *Xie et al., 2020*) revealed that TAD-like domain structures with spatially segregated heterogeneous conformations are present in single cells even without cohesin. The physical boundaries of TAD-like domains, identified from individual 3D structures, vary from cell to cell. They exhibit a preference to reside at specific genomic positions only upon averaging, as found in the Hi-C experiments. The boundary probability at each locus is the proportion of chromosome structures in which the locus is identified as a domain boundary in the 3D space. The locations of prominent peaks in the boundary probability frequently overlap with TADs detected by the population-level Hi-C maps.

To explore the relation between P-TADs in ensemble averaged contact maps and preferential boundaries in individual 3D structures of chromosomes, we first calculated individual spatial distance matrices using 10,000 simulated 3D structures that were used to identify the single-cell domain physical boundaries (*Bintu et al., 2018*). The physical domain boundaries identified from the 3D structures are the chromosome loci that spatially separate two physical clusters. It is constructed by comparing the spatial distances between a reference locus with the up- and downstream chromosome segments (*Bintu et al., 2018*). Specifically, we calculated the median values of pairwise distances between the reference loci and the upstream loci, and also the median values of pairwise distances between the reference loci and the downstream. The ratio of these two quantities is defined as boundary strength. If a locus's boundary strength is above a predefined threshold, this locus is defined as a physical boundary locus. The idea is that a physical boundary has a large ratio as it spatially separates upstream and downstream chromatin segments. Based on these boundary positions in individual cells, we define the boundary probability of a locus as the probability (fraction of all individual structures) of this locus being the physical boundary in an ensemble of individual structures. The detailed mathematical definition is provided in the section 'Boundary strength and boundary probability' and illustrated in *Appendix 5—figure 7*. We find preferential domains, with high peaks, in the boundary probability along the genomic region as well as variations in single-cell domains both in $P_L = 1$ and $P_L = 0$ (see *Appendix 5—figure 9*).

The CCM simulations show that TADs with epigenetic switches across the boundary are likely to be preserved after cohesin-associated CTCF loop loss. Furthermore, *Figure 3f* (blue dashed lines) shows that single-cell domain boundaries preferentially reside at the TAD boundaries with epigenetic switches, leading to a prominent signature for the structural ensemble after averaging over a population of cells. Interestingly, the P-TAD has prominent peaks in the boundary probabilities (in both the WT and cohesin-depleted cells), sometimes even without epigenetic switch, at the same genomic position as in the contact map (green lines in *Figure 3f and g*). These observations imply that the presence of physical boundaries in the 3D structures may be used to identify P-TADs, especially in instances when there are no epigenetic switches. We should note that the simultaneous presence of peaks in the boundary probabilities in both WT and cohesin-depleted cells is a signature of P-TADs.

Not all preferential boundaries identified in the distance matrices of the WT cells coincide with the TADs detected using the contact map (*Bintu et al., 2018*). There is discordance in the TAD boundaries and high peaks in the boundary probability. The top panel in *Figure 3h* (magenta lines) shows that in the (102–104.5) Mb range TopDom predicts that there are three P-TADs after loop loss (see the top panel with $P_L = 0$). There are two prominent peaks in the WT boundary probability whose boundaries coincide with the TADs predicted by TopDom (see the bottom panel with $P_L = 1$ in *Figure 3h*). But the peak height for the third TAD is very small. At best, one can deduce from the boundary probabilities (compare the results in *Figure 3h* for $P_L = 1$ and $P_L = 0$) that the middle TAD is preserved, which would be consistent with the TopDom prediction.

We calculated a standardized Z-score for the boundary probability in the genomic region in order to determine the preferred boundaries in single-cell domains. The number of P-TADs that are accounted for by prominent boundary peaks increases if Z-score is reduced. This implies that some P-TADs detected in the contact maps using TopDom have weak physical boundaries in the 3D structures. We considered the maxima, with Z-score values larger than 0.7, as preferred boundaries in order to determine whether P-TADs arise due to the presence of strong physical boundary. With this criterion, we obtained good agreement for the mean length of the TADs detected in the contact map using the TopDom method. The averaged sizes of the TADs in Chr13 using TopDom and boundary probability are ~565 kbs and ~535 kbs, respectively. Quantitative analysis of the boundary probabilities along the genomic region revealed ≈66% of the P-TADs in Chr10 and Chr13 have preferential positioning in single-cell domains (green bars in *Figure 3b*). Most P-TADs with epigenetic switches display prominent peaks in the boundary probabilities (≈ 85%).

The primary lessons from the simulations, which form the basis for analyzing the experiments on chromosomes from mouse liver and HCT-116 cell lines, are (1) switch in the epigenetic state across the TAD boundary is a predominant factor in determining the P-TADs after CTCF loop deletion. (2) The presence of peaks in the boundary probabilities in both the WT and cohesin-depleted cells, calculated from the 3D structures, accounts for certain fraction of P-TADs. However, in some instances TopDom predictions (used in (1)) are not compatible with boundaries deduced from 3D structures. (3) The polymer simulations show that the ensemble of 3D structures provides insights into the consequences, both at the single-cell and ensemble averaged level, of depleting cohesin.

## Structural explanation of P-TADs upon cohesin removal from analysis of Hi-C data

In order to assess whether the conclusions from simulations, summarized above, explain the experimental data, we first calculated the number of P-TADs whose boundaries have switches in the A/B epigenetic states in mouse liver and HCT-116 cell lines. We assigned chromatin state A (active, red) or B (repressive, blue) by analyzing the combinatorial patterns of histone marks using ChromHMM (*Ernst and Kellis, 2012*; see section 'Analyses of the experimental data' and *Appendix 5—figure 8*). An average over the 20 chromosomes shows that 280 P-TADs were associated with a switch between A and B epigenetic states upon ΔNipbl in mouse liver (blue bar in *Figure 4a*). The corresponding number of P-TADs, averaged over 23 chromosomes, with epigenetic switches is 396 after deleting *RAD21* in HCT-116 (blue bar in *Figure 4b*). Not unexpectedly, TADs with epigenetic switches across their boundaries are preserved with a high probability after cohesin deletion. We also find that a large number of P-TADs are accounted (green bars in *Figure 4a and b*) for by the presence of peaks in the boundary probabilities in the cohesin-depleted cells, which we discuss in detail below.

We then searched for a structural explanation for P-TADs in the two cell lines (*Figure 4a and b*). A plausible hint comes from the CCM simulations (*Figure 3*), which show that boundary probabilities, whose calculations require 3D structures, are good predictors of P-TADs. This implies that peaks in the boundary probabilities should correspond to P-TADs. Similar findings are obtained by analyzing the experimental data (*Figure 4c and d*). Additional examples are discussed in *Figure 5*. In light of these findings, we wondered whether, in general, physical boundaries can be inferred directly using Hi-C data from ensemble experiments instead of performing polymer simulations. To this end, we used the HIPPS method (see Appendix 1) to calculate an ensemble of 3D structures with the Hi-C contact map as the only input. Several conclusions follow from the results in *Figure 5*. (1) *Figure 5a and b* show that HIPPS faithfully reproduces the Hi-C contact maps. (2) Using the ensemble of 3D structures, we calculated the locus-dependent boundary probabilities for both the WT and cohesin-depleted cells.

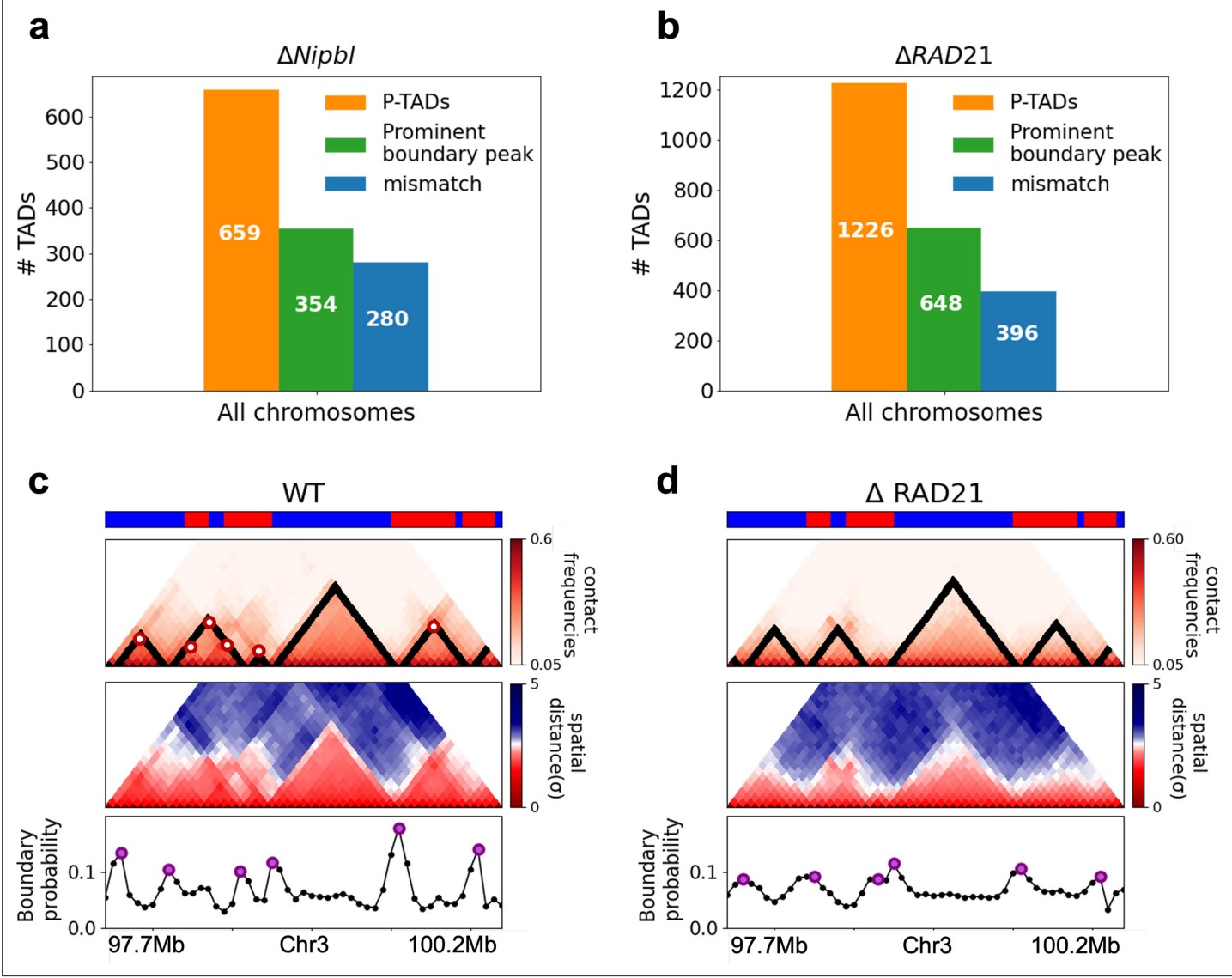

**Figure 4.** Classification of preserved topologically associating domains (P-TADs) from Hi-C maps from two cell lines and link between boundary probability peak and epigenetic switch. (**a**) The number of P-TADs in all the chromosomes (orange bar taken from *Figure 1a*) that are accounted for by epigenetic switches (blue bar) as well as peaks in the boundary probability (green bar) after *Nipbl* loss in mouse liver. (**b**) Same as (**a**) except the analyses is done using experimental data are for HCT-116 cell after Δ*RAD21*. (**c**) Example of P-TAD in the WT 97.7–100.2 Mb region of Chr3 from HCT-116 cell line. The mean distance matrices calculated using the 3D structures are shown in the middle panel. The dark-red circles at the boundaries of the TADs in the contact maps are loop anchors detected using HiCCUPS (*Durand et al., 2016*). The peaks in the boundary probability (bottom panel) are shown by purple circles. Epigenetic switch coincides with peak in the boundary probability (compare top and bottom panels). Bottom plot shows the probability for each genomic position to be a single-cell domain boundary. (**d**) Same as (**c**) except the results correspond to the absence of *RAD21*. Although not as sharp, there is discernible peak in the boundary probability when there is an epigenetic switch after removal of *RAD21*.

A comparison of the peak positions in the averaged boundary probabilities and the TAD boundaries shows that they often coincide, although there are discordances as well (*Figure 5* and *Appendix 5—figures 13 and 14*). (3) When there is a switch in the epigenetic states, a substantial fraction of P-TADs have high peaks in the boundary probabilities (see *Figure 5* and *Appendix 5—figure 13*). As in the simulations, a large fraction of P-TADs ( ≈67%) have high peaks in the boundary probabilities. Taken together, the results show that the predictions using boundary probabilities and the TopDom method are consistent. (4) Analyses of the experiments suggest that the epigenetic state as well as the presence of physical boundary in the 3D structures have to be combined in order to determine the origin of P-TADs in cohesin-depleted cells.

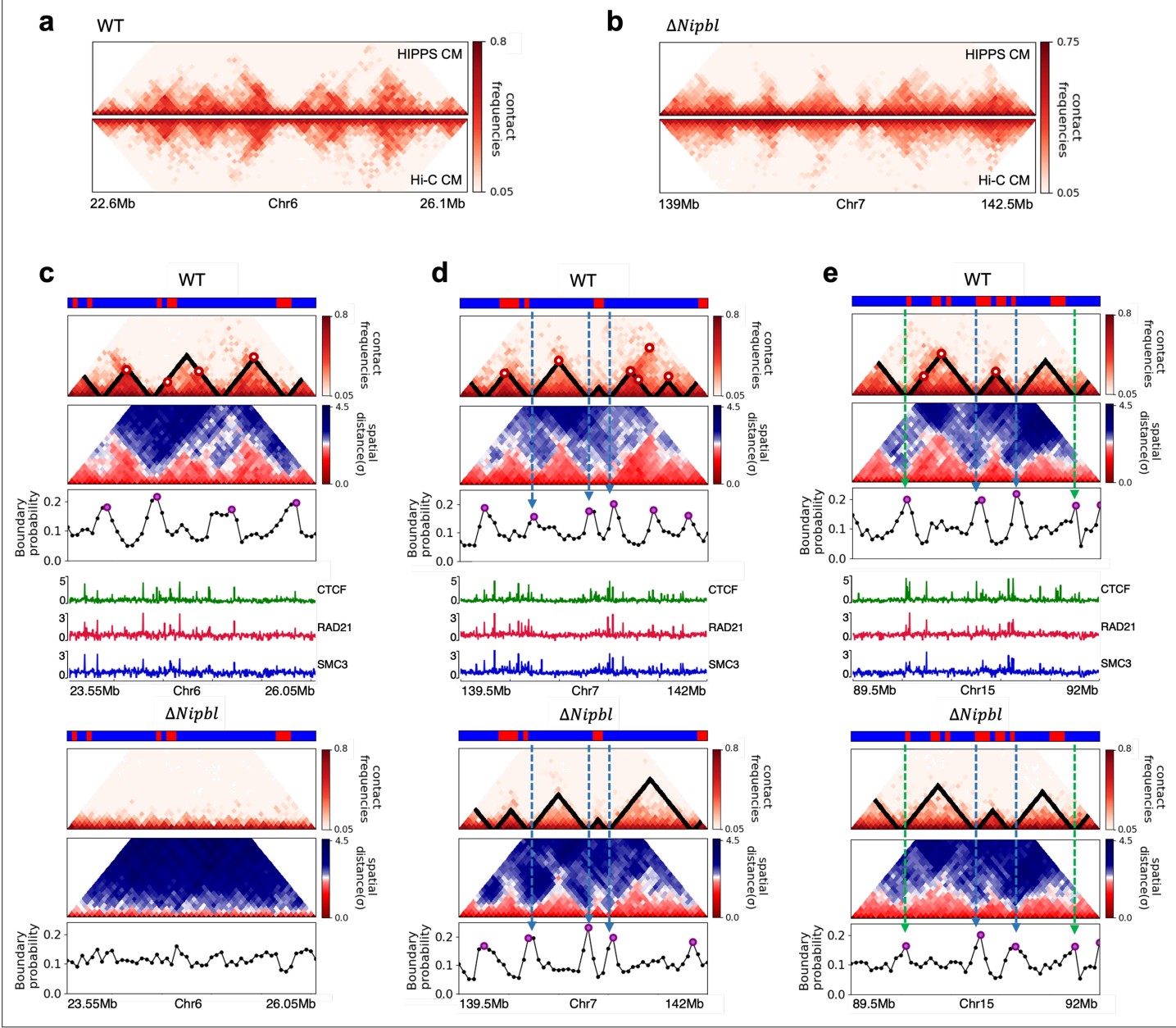

**Figure 5.** Fate of topologically associating domains (TADs) after Δ*Nipbl* in mouse liver cells. (**a, b**) Comparison between Hi-C (lower) and calculated contact maps (upper) using the 3D structures obtained from the Hi-C-polymer-physics-structures (HIPPS) method for the 3 Mb genomic regions (Chr6: 22.6–26.1 Mb in WT cells and Chr7: 139–142.5 Mb in *Nipbl*-depleted cells), respectively. The distance threshold for contact is adjusted to achieve the best agreement between HIPPS and experiments. Calculated contact maps are in very good agreement with Hi-C data for both WT and *Nipbl*-depleted cells. (**c**) Complete loss (Chr6: 23.55–26.05 Mb) of TADs in Δ*Nipbl*. (**d, e**) Preserved topologically associating domains (P-TADs) (Chr7: 139.5–142 Mb and Chr15: 89.5–92 Mb). The plots below the scale on top, identifying the epigenetic states (*Ernst and Kellis, 2012*), compare 50-kb-resolution Hi-C contact maps for the genomic regions of interest with *Nipbl* (upper) and without *Nipbl* (lower). Mean spatial distance matrices, obtained from the Hi-C contact matrices using the HIPPS method (*Shi and Thirumalai, 2021*), are below the contact maps. The dark-red circles at the boundaries of the TADs in the contact maps are loop anchors detected using HiCCUPS (*Rao et al., 2014*). ChIP-seq tracks for CTCF, RAD21, and SMC3 in the WT cells (*Schwarzer et al., 2017*) illustrate the correspondence between the locations of the most detected loop anchors and the ChIP-seq signals. Bottom plots give the probabilities that each genomic position is at a single-cell domain boundary in the specified regions. Purple circles in the boundary probability graph represent the physical boundaries. A subset of physical boundaries in P-TADs coincides with epigenetic switches (blue lines), indicating that the probabilities of contact at these boundaries are small. P-TADs in (**e**), demarcated by green lines, have high peaks in the boundary probability in the absence of epigenetic switch.

With the near quantitative agreement with experiments, we performed detailed analyses, based on the epigenetic switches and boundary probabilities for chromosomes from the mouse liver (*Figure 5*). The Appendices contain analyses of the experimental data, and the results for HCT-116 cell are given in *Appendix 5—figure 13*. To illustrate different scenarios, we consider the 2.5 Mbs regions from Chr6 (*Figure 5c*), Chr7 (*Figure 5d*), and Chr15 (*Figure 5e*). (1) For Chr 6, there are three TADs according to TopDom (*Figure 5c*) in the WT. Upon Δ*Nipbl*, these TADs are abolished (compare the top and bottom panels in *Figure 5c*). The epigenetic track indicates that the region is mostly in the repressive (B) state. Quantification of the boundary probabilities along the 2.5 Mb region of Chr6 shows that the TADs also lack physical boundaries upon Δ*Nipbl*. (2) Examples of P-TADs that satisfy the epigenetic switch criterion are shown in *Figure 5d*. Using TopDom, we identified several TADs (top panel in *Figure 5d*) in this region of Chr7. It is interesting that the boundary probabilities obtained from the HIPPS-generated distance matrices are also large when there is a switch in the epigenetic states. In these examples, both epigenetic switches and boundary probabilities give consistent results (see the dashed blue lines in *Figure 5d*). Two TADs in the WT (the ones on the right in the upper panel in *Figure 5d*) merge to form a single TAD in the Δ*Nipbl*. This observation is in accord with the expectation based on epigenetic switch, whose corollary is that if there is a TAD within a region that contains predominantly A or B type loci they ought to merge upon Δ*Nipbl*. (3) In the 2.5 Mb region of Chr15, there are three TADs in the WT (top panel in *Figure 5e*). The first and the third TADs have an epigenetic switch at only one boundary (blue dashed line), and the expectation is that they would not be preserved upon *Nipbl* removal. However, the boundary probabilities show that the TADs have physical boundaries in both, and thus they are preserved. Taken together, the results in *Figure 5* show that by combining the epigenetic switches (Hi-C data is sufficient) and the boundary probabilities (3D structures are required), one can account for a number of P-TADs.

## Single-cell structural change upon cohesin depletion

Finally, we asked whether the HIPPS method captures the 3D structural changes in cohesin-depleted cell at the single-cell level. To this end, we compare the structures obtained using HIPPS with the imaging data (*Bintu et al., 2018*), which examined the consequences of Δ*RAD21* in HCT-116. We used HIPPS, with Hi-C contact map as input on the same genomic region as in the experiment (*Bintu et al., 2018*), to generate the 3D structures. The results of our calculations for 2.5 Mbp in Chr21 (34.6–37.1 Mb) region from HCT-116 cell line for the WT and Δ*RAD21* are presented in *Figure 6*. The distance maps were calculated from the 3D structures generated using the HIPPS method (*Figure 6a*). The mean distance maps for the WT and Δ*RAD21* are shown on the left and right panels in *Figure 6b*. Similar results for Chr 4 in mouse liver cell (Chr2 in HCT-116) are displayed in *Appendix 5—figure 10* (see also *Appendix 5—figure 11*).

Several conclusions may be drawn from the results in *Figure 6*. (1) There are large variations in the distance matrices and the domain boundary locations/strengths from cell to cell (*Figure 6c and d*). This finding is in excellent agreement with imaging data (*Bintu et al., 2018*). (2) In both experiments and our calculations, there are TAD-like structures at the single level even after *RAD21* is removed (see the right panel in *Figure 6c* for the theoretical predictions). TAD-like structures in single cells with and without cohesin have also been found using the strings and binders polymer model (*Conte et al., 2020*). (3) The calculated boundary strength distribution (blue histogram in the left panel in *Figure 6d*) for the WT is in reasonable agreement with the measured distribution (purple histogram from *Bintu et al., 2018*). Similarly, the calculated and measured boundary strength distributions for Δ*RAD*21 cells are also in good agreement (right panel in *Figure 6d*). Just as in experiments (*Bintu et al., 2018*), we find that the distributions of boundary strength are the same in the WT and in cells without *RAD21*. (4) We also find that the theoretically calculated average locus-dependent boundary probability is in very good agreement with the reported experimental data (compare the curves in the left panel in *Figure 6e* for the WT and the ones on the right for Δ*RAD*21 cells).

## P-TADs are due to enhancer/promoter interactions

Cohesin is thought to directly or indirectly regulate enhancer–promoter (E–P) interactions. However, a recent Micro-C experiment discovered that E–P and promoter–promoter (P–P) interactions are, to a large extent, insensitive to acute depletion of cohesin (*Hsieh et al., 2022*). It has been previously shown that E–P/P–P interactions form one or multiple self-associating domains, strips that extend

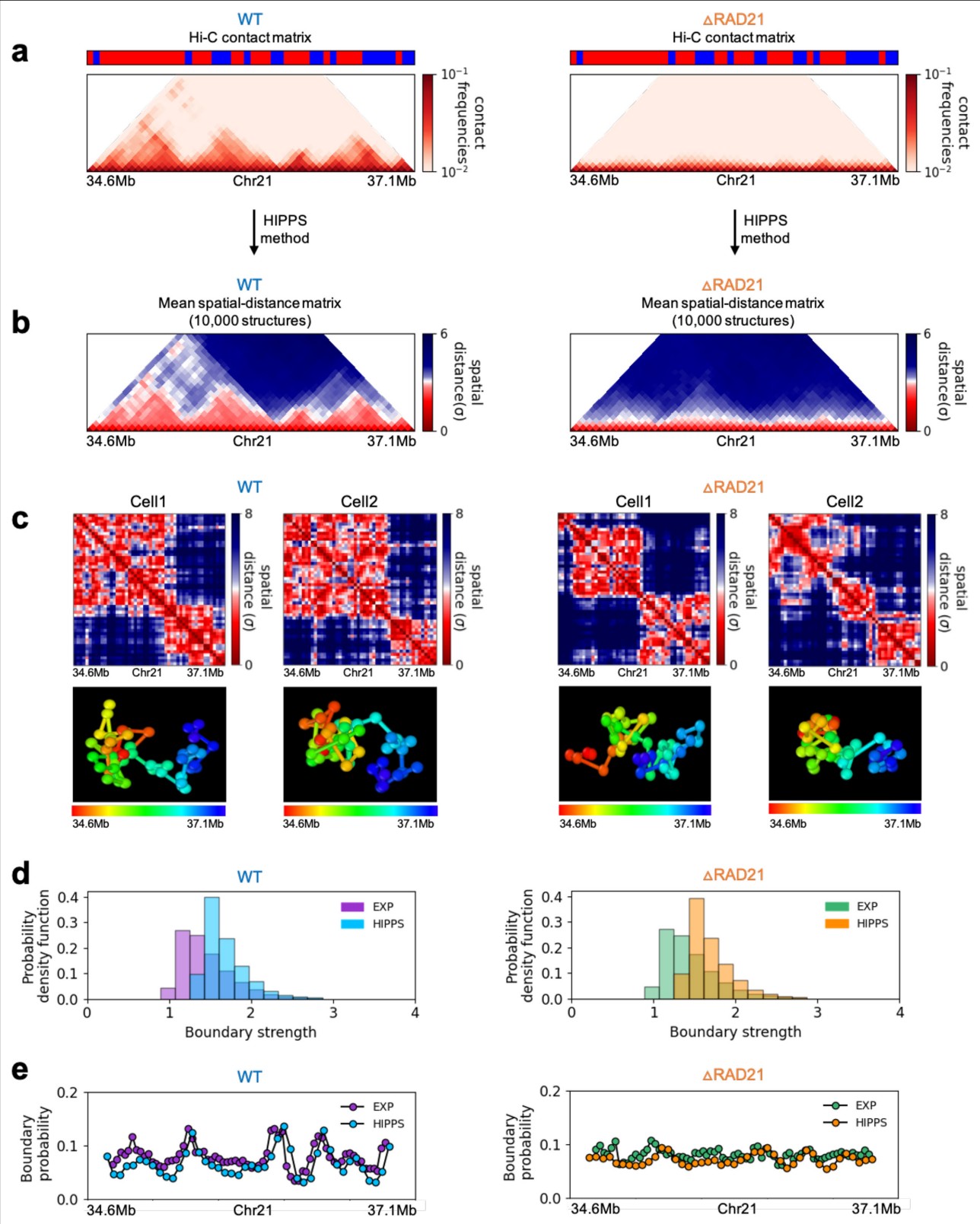

**Figure 6.** Calculated 3D structures produce topologically associating domain (TAD)-like structures found in imaging experiments. On the left panels are results for wild-type (WT) (ΔRAD21) for (Chr21: 34.6–37.1 Mb). For visualization purposes, we adopted the color scheme used in the imaging study (**Bintu et al., 2018**). (**a**) Hi-C contact maps with (left) and without RAD21 (**Rao et al., 2017**). (**b**) Mean distance matrices calculated from the Hi-C-polymer-physics-structures (HIPPS)-generated 3D structures. (**c**) Examples of calculated single-cell distance matrices with (left) and without (right)

*Figure 6 continued on next page*

*Figure 6 continued*

*RAD21*. Schematic of structures for the two cells under the two conditions are given below. (**d**) Distribution of the boundary strengths, describing the steepness in the changes in the spatial distance across the boundaries. The left is for the WT (Δ*RAD21*) cells. The blue (purple) histogram was calculated using HIPPS (experiments). (**e**) Position-dependent boundary probability for the WT (left) *RAD21*- deleted cells (right). The curve in blue (purple) is the calculated (measured) boundary probability for the WT cells. The orange (green) curve is from the calculations (experiments). The plots show that the location of prominent peaks in the calculated boundary probability is in excellent agreement with experiments for the WT cells (left panel). Without RAD21, high peaks are absent in both cases (right panel).

from domain borders and loop-like structures at their intersections at a finer scale (*Mifsud et al., 2015*; *Schoenfelder et al., 2015*; *Hsieh et al., 2020*). Inspired by the recent finding (*Hsieh et al., 2022*), we explored whether P-TADs that arise in the absence of epigenetic switches are required for the maintenance of finer-scale E–P and P–P interactions. We analyzed the Micro-C data (*Hsieh et al., 2022*) in order to shed light on this issue. The left panel in *Figure 7a* shows cohesin-associated (green dashed line) and cohesin-independent (blue dashed line) TAD structures (defined using TopDom) in the WT cells. In the latter case, the E–P and P–P loops (blue circles) are at the boundary of the TADs even in the absence of *epigenetic switch*, implying that it is a domain that is needed for E–P or P–P communication. Interestingly, the TADs were also conserved upon cohesin loss (right panel in *Figure 7a*). Analyses of the 3D structures (*Figure 7b*) reveal that the TADs with E–P/P–P loops have strong physical boundaries *sans* cohesin. *Figure 7c* shows an example of a TAD with both E–P/P–P loops and cohesin/CTCF loops at the boundary in the WT cells that is retained after cohesin deletion and is associated with prominent boundary peaks. We propose that only a subset of TADs is conserved, potentially for functional reasons.

The statistical analyses of all the P-TADs observed in the Micro-C contact maps across all the chromosomes show that 525 out of 1536 P-TADs have E–P/P–P loops that coincide with their boundaries (*Figure 8*). Taken together, our observations suggest that the maintenance of E–P/–P–P interactions could be the origin of the P-TADs even if there are no epigenetic switches. It is worth emphasizing that these conclusions can only be obtained by analyzing the 3D structures, which we calculated from the Micro-C contact maps using the HIPPS method (*Shi and Thirumalai, 2021*) that does not rely on polymer simulations.

## Discussion

By analyzing the experimental Hi-C data (*Rao et al., 2017*; *Schwarzer et al., 2017*), we first showed that upon cohesin loss a non-negligible fraction of TADs is preserved, which was not previously noticed. To examine the factors that control the P-TADs, we then performed polymer simulations of two chromosomes in the presence and absence of cohesin-associated CTCF loops from the GM12878 cell line. The polymer simulation results were used to generate the hypotheses for the emergence of P-TADs, which were used to explain the major findings reported in the experiments in mouse liver and HCT-116 cells (*Rao et al., 2017*; *Schwarzer et al., 2017*). The simulations showed that switches in the epigenetic states across the TAD boundary account for a large fraction of P-TADs. Even in the absence of epigenetic switches, P-TADs could be preserved, as revealed by the presence of physical boundaries in the 3D structures.

Rather than performing a number of time-consuming polymer simulations, we used the data-driven approach (*Shi and Thirumalai, 2021*) that generates three-dimensional chromosome structures rapidly and accurately using the Hi-C contact maps as input. Analyses of the calculated structures, with and without *Nipbl* or *RAD21*, showed that A-type loci form larger spatial clusters after cohesin removal, consistent with the enhancement of compartments inferred from Hi-C contact maps (*Appendix 5— figures 4 and 5*). Most of the P-TADs, with epigenetic switches in the contact maps, have prominent peaks in the boundary probabilities in both WT and cohesin-depleted cells. An important conclusion from this striking finding is that not only micro-phase separation on the larger scale, $L_C$, but also some special TADs on a shorter scale, $L_{TAD}$ are encoded in the epigenetic sequence.

Remarkably, the conclusion that there are cell-to-cell variations in the distance maps noted in imaging experiments (*Bintu et al., 2018*) is affirmed in the calculated 3D structures. This finding is significant because (1) only a limited number of loci can be directly imaged whereas Hi-C data can be

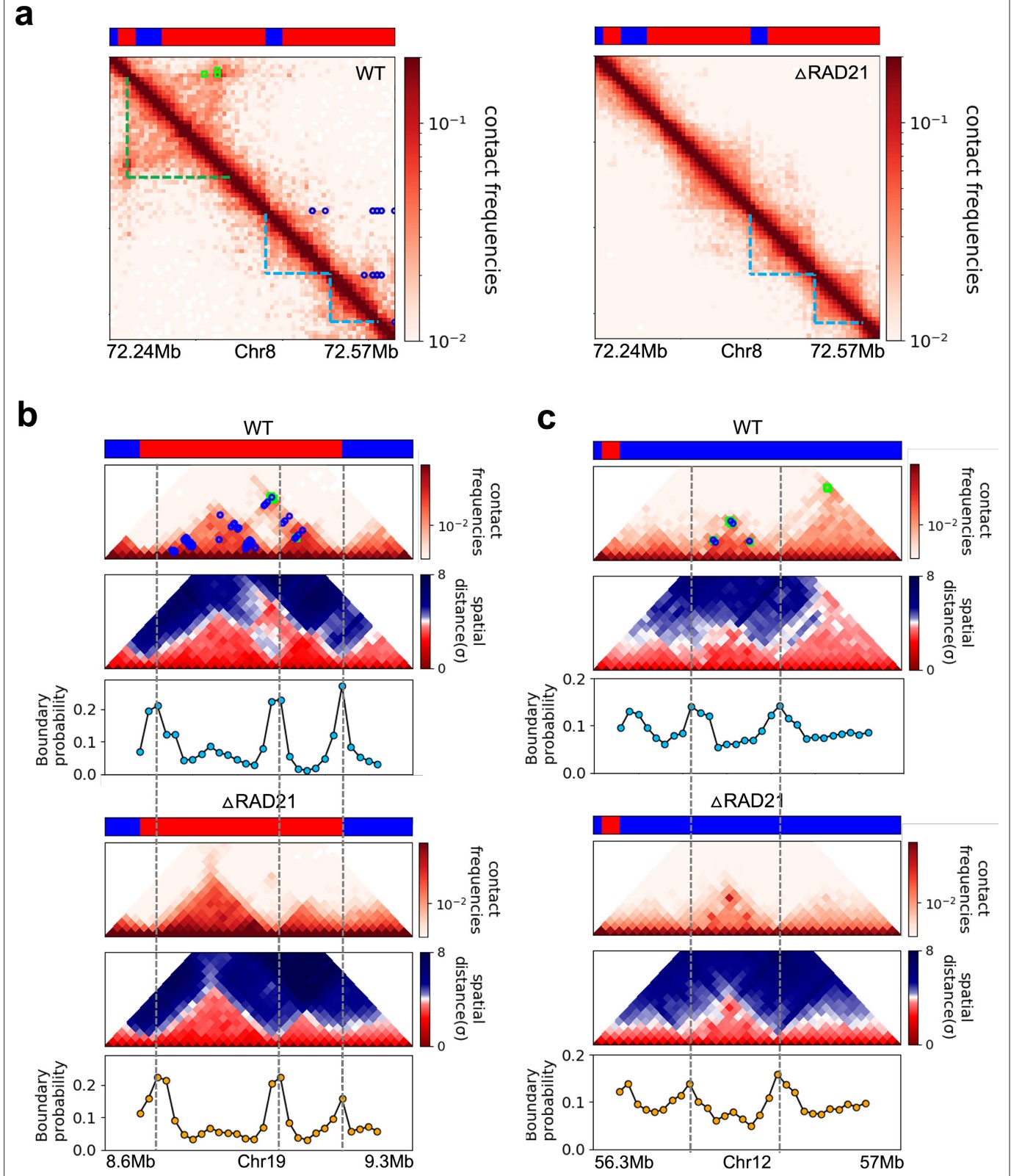

**Figure 7.** Certain topologically associating domains (TADs) enriched in enhancer–promoter/promoter–promoter (E–P/P–P) interactions at the boundary are preserved upon cohesin deletion. (**a**) Comparison between 5 kb Micro-C contact maps in the region (Chr8: 72.24–72.57 Mb) for the wild-type (WT) (left panel) and cohesin-depleted (right panel) mouse embryonic stem cells (mESC) cells (*Hsieh et al., 2022*). Location of cohesin loops (green square) and E–P/P–P (blue circles) plotted in the WT contact maps are from experiments (*Hsieh et al., 2022*). Bars above the contact map show epigenetic

*Figure 7 continued*

states (red: active; blue: inactive) annotated based on ChromHMM results (*Pintacuda et al., 2017*). The cohesin-dependent (green dashed lines) and independent (blue dashed lines) TADs were detected in the WT cells using the TopDom method with default parameter ($w = 5$). P-TADs (blue dashed lines) are also found in cohesin-deleted cells. (**b, c**) Comparison between 20 kb Micro-C contact maps and mean distance maps spanning the regions, Chr19: 8.66–9.2 Mb and Chr12: 56.4–56.9 Mb, respectively, in the presence (upper) and absence (lower) of cohesin. Bottom graph, below the distance maps, shows the boundary probability calculated from 10,000 3D structures. P-TADs between gray dashed lines were detected using the TopDom method ($w = 5$). A P-TAD with high boundary peak, without epigenetic switches, is enriched due to E–P/P–P interactions at the boundaries.

routinely generated at higher resolution, and (2) the number of Hi-C data on various cell types and species currently is far greater than that obtained from imaging data.

Let us summarize the novel results, which sets our work apart from previous insightful studies (*Rao et al., 2017*; *Schwarzer et al., 2017*; *Nuebler et al., 2018*). (1) We showed by analyzing the Hi-C data for mouse liver and HCT-116 cell lines that a non-negligible fraction of TADs is preserved, which set in motion our detailed investigations. (2) Then, using polymer simulations on a different cell type (GM12878), we generated quantitative insights (epigenetic switches as well as structural basis) for the preservation of TADs. Although not emphasized, we showed that deletion of cohesin in the GM12878 cell line also leads to P-TADs, a prediction that suggests that P-TADs may be 'universal'. (3) Rather than performing time-consuming polymer simulations, we calculated 3D structures directly from Hi-C data for the mouse liver and HCT-116 cell lines, which provided a structural basis for TAD preservation. (4) The 3D structures also showed how TAD-like features appear at the single-cell level, which is in accord with imaging experiments (*Bintu et al., 2018*). (5) Finally, we suggest that P-TADs may be linked to the maintenance of enhancer–promoter and promoter–promoter interactions by calculating the 3D structures using the recent Micro-C data (*Hsieh et al., 2022*).

## Comments on the methods

In order to explore the factors that control the P-TADs, there are two assumptions. (1) The results of the Hi-C experiments are taken at face value in the sense. We view this as an assumption because errors in the Hi-C readouts may be difficult to evaluate even though such experiments are invaluable (*Akgol Oksuz et al., 2021*). (2) The TADs were identified using TopDom, one of many TAD callers. A recent survey (*Zufferey et al., 2018*) shows that, although the finding that TADs are hierarchically organized domains is robust, there are substantial variations in the identification of these domains predicted by different methods. Although TopDom fairs reasonably well in comparison to other methods, there is no guarantee that it identifies the TAD location or the number of TADs accurately. It is only for convenience that we used TopDom as the reference to which the results using the boundary probabilities are compared. (3) Because the prediction of 3D structures using the HIPPS method does not require extensive polymer simulations, it can be used to predict the structural changes for chromosomes that are subject to large-scale perturbations. The excellent agreement between the HIPPS calculations and imaging experiments further bolsters the power of our approach.

## Methods

We performed polymer simulations for the following reasons. (1) Because all TAD-calling schemes are approximate, we evaluated the accuracy of the given protocol (TopDom [*Shin et al., 2016*] in our study) using the well-calibrated CCM. TAD identification in the CCM simulations could be made directly from the 3D structures, thus allowing us to test the validity of the TopDom method. (2) The combination of 3D structures, assignment of epigenetic states using ChromHMM (*Ernst and Kellis, 2012*), and accurate calculation of the Hi-C maps using the CCM were used to determine the origin of P-TADs. (3) An added bonus is that the polymer simulations on an entirely different cell type (human GM12878) could be used to assess the robustness of the conclusion that a certain fraction of TADs is preserved upon deletion of cohesin.

To avoid biases in the formulation of the hypothesis to explain TAD preservation, we simulated chromosomes from the human cell line (GM12878), which is different from the cell lines used in experiments (*Rao et al., 2017*; *Schwarzer et al., 2017*). In the main text, we report the results for Chr13 (19–115.10 Mbp). The total number of 50kb loci is $N = 1923$, and the total number of loop anchor pairs is 72. To ensure that the results are robust, we also simulated Chr10 (*Appendix 5—figure 3*).

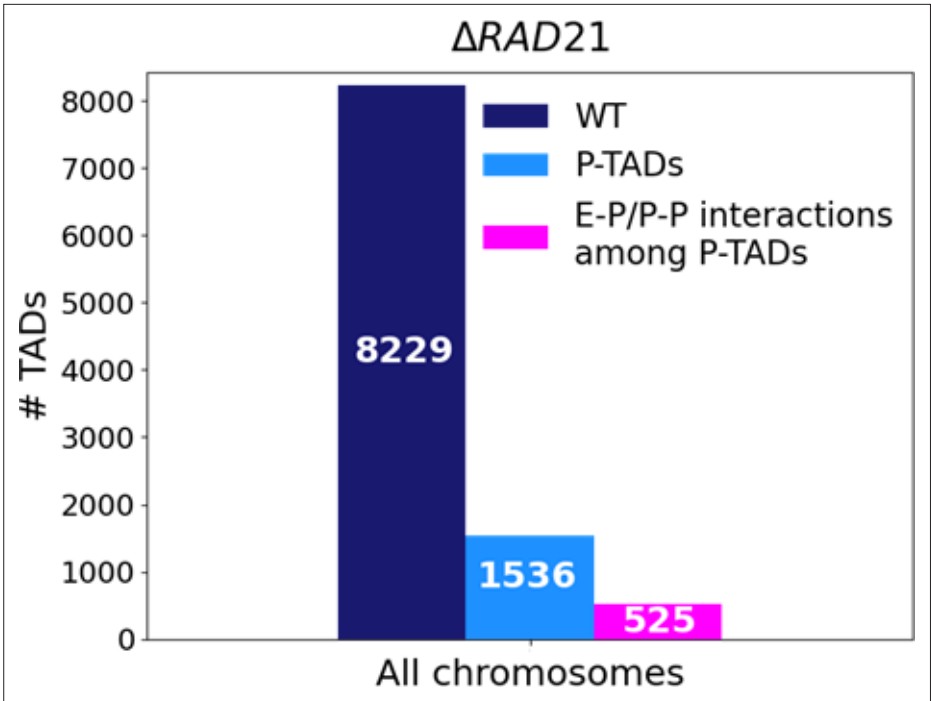

**Figure 8.** Statistics of the topologically associating domains (TADs) in chromosomes upon cohesin loss using Micro-C contact data. The number of TADs in all the chromosomes in the wild-type (WT, dark blue bar), total number of preserved TADs (P-TADs, light blue bar) after deleting RAD21, and number of P-TADs whose boundaries coincide with enhancer–promoter/promoter–promoter (E–P/P–P) interactions (magenta bar) in mESC. About a third of the P-TADs are associated with E–P/P–P interactions.

## CCM for chromosomes

We modified the CCM (*Shi et al., 2018*) in order to simulate full-length interphase chromosomes. In the CCM, chromosomes are modeled as a self-avoiding copolymer, with A (B)-type loci representing the active (repressive) epigenetic state. The connectivity between two nearest-neighbor loci ($nn$), $i$ and $i + 1$, separated by a distance $r_{nn} = |r_i - r_{i+1}|$, is given by a finitely extensible nonlinear elastic (FENE) potential,

$$U^B_{FENE}(r = r_{nn}) = -\frac{1}{2}K_{FENE}R_0^2 \ln\left[1 - \left(\frac{r}{R_0}\right)^2\right], \tag{1}$$

where $K_{FENE}$ is the spring constant and $R_0$ is an estimate of the equilibrium bond distance. Non-bonded interaction between two loci that are not directly connected to each other is given by the Lennard–Jones (LJ) potential,

$$U_{LJ}(r_{\alpha\beta} = r = |r_i - r_j|) = 4\epsilon_{\alpha\beta}\left[\left(\frac{\sigma}{r}\right)^{12} - \left(\frac{\sigma}{r}\right)^6\right], \tag{2}$$

where $\alpha$ and $\beta$ can be either A or B. Finally, we used a harmonic potential for the CTCF loop anchors $p$ and $q$ that are typically stabilized by cohesin. The loop anchor potential is,

$$U^L_h(r = |r_p - r_q|) = K_h(r - r_{0,h})^2, \tag{3}$$

where $K_h$ is the spring constant and $r_{0,h}$ is the equilibrium length between the CTCF loop anchors. The CCM energy function is

$$U_{CCM} = \sum_{i=1}^{N-1} U^B_{FENE}(r_{i,i+1}) + \sum_{i=1}^{N-1}\sum_{j=i+1}^{N} U_{LJ}(r_{i,j}) + \sum_{\{p,q\}} U^L_h(r_{p,q}) \tag{4}$$

**Table 1.** Parameters for bonding potentials.

| $K_{FENE}/k_BT\sigma_{50k}^{-2}$ | $r_0/\sigma_{50k}$ | $K_h/k_BT\sigma_{50k}^{-2}$ | $r_{0,h}/\sigma_{50k}$ |
|---|---|---|---|
| 2.497 | 5.199 | 24.97 | 3.916 |

The unit of energy is $k_BT$, where $k_B$ is the Boltzmann constant and $T$ is the temperature.

We used the CCM simulations in order to deduce the mechanisms for preservation of certain TADs when the loop anchors are deleted. The simulations must reproduce the two major findings (*Rao et al., 2017*; *Schwarzer et al., 2017*): (1) propensity of the A and B loci to segregate should be enhanced upon removal of cohesin; and (b) a fraction of TADs should be preserved upon cohesin or cohesin-associated CTCF loop loss. Each locus in the polymer is either A type (active locus is in red) or B type (repressive locus is in blue) (*Appendix 5—figure 2a*). The locus type is determined using the Broad ChromHMM track (*Rosenbloom et al., 2013*; *Ernst and Kellis, 2010*; *Ernst et al., 2011*). There are 15 chromatin states, out of which the first 11 are related to gene activity, based on which we group states 1–11 as active state (A) and states 12–15 as repressive state (B). The locations of the CTCF/cohesin-mediated loop anchors, which are fixed in the polymer simulations (cohesin is present), are obtained from the Hi-C data (*Rao et al., 2014*) (GSE63525_GM12878_primary+replicate_HiCCUPS_with_motifs.txt.gz). Removal of the loop constraints mimics the absence of cohesin. In the WT simulations, the probability of loop anchor, $P_L$, being present is unity. To model cohesin depletion, we set $P_L = 0$ to assess the impact of deleting the loops on compartments and TADs.

## CCM at 50kb resolution

In our previous study (*Shi et al., 2018*), we used 1200 bps resolution. Here, we used 50,000 bps (50kb) resolution in order to model the entire length of the chromosomes. To determine the size of each locus, with $N_{bp}$ base pairs, we assume (*Rubinstein and Colby, 2003*) that the radius of gyration is $R_g \sim N_{bp}^{1/3}$. Assuming that a locus, with $\sigma_{1,200}$ and $\sigma_{50k}$, represents a condensed polymer, with 1.2k and 50k base pairs, respectively, we expect that $R_{g,1200} \sim (1,200)^{1/3}$ and $R_{g,50,000}$. By using this relation, we estimated the size of each locus, $\sigma_{50k} = 3.466\sigma_{1200kb}$. Similarly, the mass of the locus at 50 kb resolution is modified as $m_{50k} = m_{1,200} = 41.66\, m_{1,200}$, where $m_{1,200} = 1$. The parameters for the bonding potentials (*Equations 1* and *3*) at 50 kbps resolution of the CCM shown in *Table 1*.

## Effective energy scales

The creation of CCM was motivated by the experimental observation that active and repressive loci segregate on a few megabase scale. By adopting the Flory–Huggins (FH) theory (*Rubinstein and Colby, 2003*), the spatial segregation between A and B loci is modeled using a weaker A–B attraction compared to A–A and B–B interactions. With the assumption that $\epsilon_{AA} = \epsilon_{BB} = \epsilon$, which is made for simplicity, the only free parameter in the CCM is $\epsilon_{AB}$. By fixing the ratio $\frac{\epsilon}{\epsilon_{AB}}$ to $\frac{11}{9}$, the simulated contact maps are in reasonable agreement with the Hi-C maps. Although a large number of energy functions could reproduce the Hi-C map (*Di Pierro et al., 2016*; *Falk et al., 2019*; *Jost et al., 2014*), the CCM is perhaps the simplest copolymer model with only one unknown energy parameter, $\epsilon$.

## Simulations

We performed Langevin dynamics (LD) simulations using the LAMMPS simulator by integrating the equations of motion,

$$m\frac{d^2r_i}{dt^2} = -\nabla_{r_i}U - \zeta\frac{dr_i}{dt} + \delta F_i(t), \tag{5}$$

where $r_i$ is the position vector of the $i$th locus and $-\nabla_{r_i}U$ is the force on the $i$th locus, and $\zeta$ is the friction coefficient that is chosen to be $\frac{0.01m_{50k}}{\tau_{50k}}$. The random force $\delta F_i(t)$ satisfies $\langle\delta F_i(t)\rangle = 0$, $\langle\delta F_i(t) \cdot \delta F_i(t')\rangle = 6\zeta k_BT\delta(t - t')$. We first did simulations using a small time step, $10^{-6}\tau_{50k}$, with only repulsive pairwise interactions between the loci to avoid numerical instabilities. After a certain number of time steps, the loci associated with the loops are in proximity and undergo fluctuations around their equilibrium bond distance. At this stage, we increased the time step to $10^{-2}\tau_{50k}$, turned on the attractive pairwise interactions, and continued the simulations for $10^8\Delta t_{50k}$. We then performed LD

simulations for an additional $10^8 \Delta t_{50k}$ to compute the structural properties. Because we are only interested in equilibrium structures, the values of $\frac{m}{\tau}$ and $\zeta$ are irrelevant.

## Contact map

We calculated the contact frequencies, $C_{ij}$, between loci $i$ and $j$ by computing the distance, $r_{ij} = |r_j - r_i|$, between them and counting the number of instances when $r_{ij} < 1.75\sigma_{50k}$. The set of elements, $C_{ij}$, constituting the contact map is a 2D representation of the chromosome organization (**Appendix 5—figures 2b** and **3a**).

## Pearson correlation map

To assess the accuracy of the CCM predictions, we calculated the Pearson correlation maps (**Appendix 5—figures 2c** and **3b**) by first transforming the simulated contact maps and the Hi-C data to a $log_e$ scale. For each element, $C_{ij}$, we calculated, $Z_{ij}$, using

$$Z_{ij} = \frac{\left(C_{ij} - \langle C_s \rangle\right)}{\sigma_s}, \tag{6}$$

where $\langle C_s \rangle = (1/(N-s))\sum_{i<j}\delta(s-(j-i))C_{ij}$, and $\sigma_s$ is the standard deviation associated with $C_s$. The Pearson correlation coeficient (PCC), $\rho_{ij}$, is calculated between the $i$th row, $X_i$, and the $j$th column, $Y_j$, associated with the matrix $Z$ whose elements are $Z_{ij}$. The PCC is $\rho_{X,Y} = \frac{E(X-\mu_X)(Y-\mu_Y)}{\sigma_X \sigma_Y}$, where $E$ denotes expectation, $\mu_X$ and $\mu_Y$ are the means of $X$ and $Y$, respectively, and $\sigma_X$ ($\sigma_Y$) is the standard deviation of $X$ ($Y$).

## Kullback–Leibler (KL) divergence

To measure the difference between two probability distributions that are functions of the same variable $x$, we calculated the KL divergence, $D_{KL}(p(x), q(x))$, which is a measure of the information loss when $q(x)$ is used to approximate $p(x)$. Here, $p(x)$ and $q(x)$ are the two probability distributions of a discrete random variable $x$. Using the KL divergence, the difference between the PCC probability distributions obtained from the simulations, $p^{CCM}$, and experiments, $p^{EXP}$, was calculated. We define $D_{KL}(p^{EXP}, p^{CCM})$ as $\sum_{i,j} p_{ij}^{EXP} log(p_{ij}^{EXP}/p_{ij}^{CCM})$ as shown in **Appendix 5—figure 2d** and **Appendix 5—figure 3c**.

## Ward linkage matrix (WLM)

We used the WLM, an agglomerative clustering algorithm method, to reveal the hierarchical organization on different length scales (**Appendix 5—figures 2h** and **3f**). In our previous study (**Shi et al., 2018**), we showed that the contact probability is inversely proportional to a power of the spatial distance, $P_{ij} \propto R_{ij}^{-4.1}$. This relationship provides a way to convert Hi-C contact matrix into spatial distance matrix. We computed WLM with our simulated spatial distance matrix, which is directly calculated in the simulations. To compare with experiments, we converted Hi-C contact matrix into spatial distance matrix, $\mathbb{R}_{\text{exp}}$, using the relation $R_{ij} \propto P_{ij}^{-1/4.1}$. The WLM, $\mathbb{W}$, from $\mathbb{R}_{\text{exp}}$ and simulated spatial distance matrix, $D = 3$, can be calculated as described previously (**Shi et al., 2018**).

## Density-based spatial clustering of applications with noise (DBSCAN)

DBSCAN is a clustering algorithm (**Sander et al., 1998**) that finds regions of high density by grouping together data points that are in proximity based on spatial distribution. For DBSCAN, two parameters, *Epsilon* and *MinPoints*, are required; *Epsilon* is a threshold distance between two loci that is used to classify whether they belong to the same cluster, whereas *MinPoints* is the minimum number of data points needed to form a dense region. The *MinPoints* can be derived from the spatial dimensions, $D$, in the data points, as *MinPoints*. We use the recommended value (**Sander et al., 1998**) *MinPoints* = 2 × $D$ with $D = 3$.

The optimal *Epsilon* value is determined using $k$-distance graph. We set *MinPoints* = 6 and calculated the distance from every point to the $k$th nearest neighbor in each cell. The $k$-distances are plotted in an ascending order, and a reasonable value corresponds to the maximum curvature (elbow) in this plot. It is likely that optimal values depend on the chromosomes, A/B loci type and the cell type. For example, we found that the optimal *Epsilon* values are 1.7 (1.15)$\sigma_{CCM,Chr13}$, 1.0 (0.8) $\sigma_{mouseliver,Chr19}$,

and 1.6 (1.4)$\sigma_{HCT116,Chr15}$ for A (B) loci in Chr13 (CCM), Chr19 (mouse liver), and Chr15 (HCT-116) of both WT and CTCF loops/cohesin-depleted cells, respectively (each $\sigma$ represents the average distance between $i$ and $i + 1$ loci in the chromosome). With the optimal parameters, we identified the number of A (B) clusters, $N_A(N_B)$, in 10,000 individual structures in the chromosome (see **Appendix 5—figures 4 and 5**). In addition, we calculated the size of each cluster, $S_A(S_B)$, which is defined as (the number of A (B) loci within the cluster)/(the total number of A(B) loci within the chromosome).

The compartmental strength is enhanced after the removal of CTCF loops (**Appendix 5—figure 6**), indicating that CTCF loop loss leads to an enhanced tendency for micro-phase separation (**Rao et al., 2017**; **Schwarzer et al., 2017**; **Nuebler et al., 2018**; **Wutz et al., 2017**; **Haarhuis et al., 2017**; **Sofueva et al., 2013**; **Bintu et al., 2018**; **Seitan et al., 2013**). Thus, DBSCAN, a method that relies on 3D structures, and a method that uses only the contact map produces a qualitatively consistent picture of strengthening of compartments upon cohesin loss.

## TAD and P-TAD identification

TopDom (**Shin et al., 2016**) is one of many methods used to identify TADs. The average contact frequency around each locus, $i$, between upstream ($i$ - $w$ + 1, $i$-$w$, …, $i$) and downstream ($i$ + 1, $i$ + 2, …, $i$ + $w$) regions with the free parameter, $w$, is calculated as the value of the binSignal. TAD boundaries correspond to local minima in the binSignal. Subsequently, false detections in the local minima are filtered by using the Wilcox rank sum. We used the software package and source codes of TopDom (https://github.com/jasminezhoulab/TopDom, **Bengtsson et al., 2020**) with default parameter, $w = 5$. Two aspects concerning the implementation of TopDom should be kept in mind. (1) TopDom results change depending on parameter values. Large $w$ produces big domains, reducing the total number of detected domains. (2) There are some matrix columns/rows whose contact frequencies sum up to zero. We refer to them as missing bins. We selected only the domains whose boundaries have zero or one missing bin in a 250 kb range since the presence of the missing bin influences contact insulation.

For completeness, let us define P-TADs. We detected TADs using TopDom (**Shin et al., 2016**) based on the Hi-C data. First, *P-TADs* are those that remain in *both* the WT cells and cohesin-depleted cells (**Figure 2**). If the boundaries between two TADs in cohesin-depleted cells are within ±50 kb window from the corresponding boundary in the WT, and if there is ≥80% overlap between the WT and cohesin-depleted cells, such a TAD is classified as a P-TAD. Second, *epigenetic switches across TAD boundaries* refer to the alteration of epigenetic state upon going from one TAD to the neighboring TAD (see **Appendix 5—figure 1**). For instance, one TAD consisting of predominantly euchromatin loci with the adjacent TAD comprising largely heterochromatin loci would create epigenetic switches across the boundary. We also used 3D structures of chromosomes, with and without cohesin, to calculate boundaries to determine the structural origin of P-TADs.

## P-TADs with epigenetic switches

The procedure for determining epigenetic switches is schematically shown in **Appendix 5—figure 1**. Switches that occupy only one locus (50 kb) were excluded (see *I* in **Appendix 5—figure 1**). We considered the P-TAD boundary and epigenetic switch as overlapping if they are less than 100 kb apart (*II* in **Appendix 5—figure 1**). Finally, P-TADs with epigenetic switches, consisting of <70% of sequences in identical epigenetic states, with epigenetic switches, were filtered out (*III* in **Appendix 5—figure 1**).

## Boundary strength and boundary probability

To measure the boundary strength (**Bintu et al., 2018**; **Cheng et al., 2021**) for each locus $i$, we first calculated the median distance values ($L$) of the left three columns, each extending 6-elements below the diagonal, and the median value ($R$) of the right three 6-element columns below the diagonal. Similarly, the median value ($B$) of the right three 6-element columns above the diagonal and the median value ($T$) of the left three 6-element columns above the diagonal were calculated.

The two boundary strengths, $L/R$ (start-of-domain boundary strength) and $B/T$ (end-of domain boundary strength), are computed as defined in **Appendix 5—figure 7**. The local maxima above a defined threshold in the start/end of domain boundary strengths are identified as the start/end positions of the domain boundary, respectively. This is physically reasonable because at the boundary between two TADs $\langle p_{ij} \rangle$ is low, which implies that $\langle r_{ij} \rangle$ has to be large. Based on the boundary positions in individual cells, we compute the start/end boundary probability for each locus as the fraction

of chromosomes in which the corresponding locus is identified as a start/end boundary of a domain. The average of these start and end boundary probabilities for each locus is defined as the boundary probability at the locus (*Appendix 5—figure 7*).

## Acknowledgements

We are grateful to Alistair Boettiger for discussion and useful comments. We thank Sucheol Shin, Atreya Dey, and Debayan Chakraborty for their useful discussion. This work was supported by the National Science Foundation (CHE 2320256) and the Welch Foundation (F-0019) through the Collie-Welch Regents Chair.

## Additional information

### Funding

| Funder | Grant reference number | Author |
| --- | --- | --- |
| National Science Foundation | CHE 2320256 | D Thirumalai |
| Welch Foundation | F-0019 | D Thirumalai |

The funders had no role in study design, data collection and interpretation, or the decision to submit the work for publication.

### Author contributions

Davin Jeong, Software, Formal analysis, Validation, Visualization, Methodology, Writing – original draft; Guang Shi, Software, Formal analysis, Validation, Methodology, Writing – review and editing; Xin Li, Formal analysis, Validation, Investigation, Writing – review and editing; D Thirumalai, Conceptualization, Formal analysis, Supervision, Funding acquisition, Investigation, Methodology, Writing – original draft, Project administration, Writing – review and editing

### Author ORCIDs

Davin Jeong http://orcid.org/0000-0003-2537-9363
Guang Shi http://orcid.org/0000-0002-4377-7210
Xin Li http://orcid.org/0000-0002-2510-2236
D Thirumalai https://orcid.org/0000-0003-1801-5924

Reviewer #1 (Public Review): https://doi.org/10.7554/eLife.88564.3.sa1
Reviewer #2 (Public Review): https://doi.org/10.7554/eLife.88564.3.sa2
Reviewer #3 (Public Review): https://doi.org/10.7554/eLife.88564.3.sa3
Author Response https://doi.org/10.7554/eLife.88564.3.sa4

## Additional files

### Supplementary files

• MDAR checklist

### Data availability

Since the current manuscript is a computational study, no data have been newly generated in this manuscript. For chromosome copolymer model (CCM), the molecular dynamics simulations were performed using LAMMPS (*Thompson et al., 2022*). The code for HIPPS has been uploaded and deposited on GitHub at https://github.com/anyuzx/HIPPS-DIMES (copy archived at *Shi, 2023*).

The following previously published datasets were used:

| Author(s) | Year | Dataset title | Dataset URL | Database and Identifier |
|---|---|---|---|---|
| Suhas SPR, Huang SC, St Hilaire BG, Engreitz JM, Perez EM, Kieffer-Kwon KR, Sanborn AL | 2017 | Cohesin Loss Eliminates All Loop Domains | https://www.ncbi.nlm.nih.gov/geo/query/acc.cgi?acc=GSE104334 | NCBI Gene Expression Omnibus, GSE104334 |
| Wibke S, Abdennur N, Goloborodko A, Pekowska A, Fudenberg G, Loe-Mie Y, Fonseca NA | 2017 | Two independent modes of chromosome organization are revealed by cohesin removal | https://www.ncbi.nlm.nih.gov/geo/query/acc.cgi?acc=GSE93431 | NCBI Gene Expression Omnibus, GSE93431 |
| Hsieh THS, Cattoglio C, Slobodyanyuk E, Hansen AS, Darzacq X, Tjian R | 2021 | Enhancer-promoter interactions and transcription are maintained upon acute loss of CTCF, cohesin, WAPL, and YY1 | https://www.ncbi.nlm.nih.gov/geo/query/acc.cgi?acc=GSE178982 | NCBI Gene Expression Omnibus, GSE178982 |

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

## Appendix 1

### Analyses of the experimental data

A hypothesis that emerges from the CCM simulations is that the TADs, which are preserved with high probability upon cohesin deletion, have epigenetic switches across the TAD boundary and are often accompanied by peaks in the boundary probability. In order to test this hypothesis, we analyzed the Hi-C contact map from *Schwarzer et al., 2017* (mouse liver) and *Rao et al., 2017* (HCT-116). In each experiment, cohesin loading factor, *Nipbl*, and a core component of the cohesin complex, *RAD21*, were depleted by employing a liver-specific, tamoxifen-inducible Cre driver and an auxin-inducible degron (AID), respectively. The availability of the WT and cohesin-depleted (Δ*Nipbl* or Δ*RAD21*) contact maps allows us to test the hypothesis derived from simulations.

In order to analyze the mouse liver data, we used the WT and *Nipbl*-depleted Hi-C contact maps at 50 kb resolution from GEO: GSE93431. The locations of the CTCF loops were determined using the HiCCUPS method in HiCPeaks (*Rao et al., 2014*; *Rao et al., 2022*; https://github.com/XiaoTaoWang/HiCPeaks). HiCCUPS examines each pixel in a Hi-C contact matrix and detects loops by finding the pixels that are enriched relative to local neighborhoods (pixels to its lower-left, pixels to its left and right, pixels above and below, and pixels within a doughnut-shaped region surrounding the pixel of interest). We obtained the locations of the 3301 loop anchors for 20 chromosomes from the WT Hi-C contact maps. Epigenetic landscape is determined using the Broad ChromHMM track for mouse liver (*Bogu et al., 2015*; *Bogu, 2013*; https://github.com/gireeshkbogu/chromatin_states_chromHMM_mm9). Among the 15 chromatin states in the track, we assigned states 1–10 and 15 to be in the active state (A) because they are related to gene transcription. States 11–14 correspond to heterochromatin, and hence are taken to the repressive (B).

For the HCT-116 cell, we used 50 kb resolution Hi-C map for untreated and *RAD21*-depleted cells obtained from GEO: GSE104334. Chromatin state characterization was performed using ChromHMM (*Ernst and Kellis, 2012*) considering 12 histone modifications ChIP-seq data (H3K27ac, H3K9ac, H3K27me3, H3K36me3, H3K79me2, H3K9me2, H3K9me3, H4K20me1, H3K4me1, H3K4me2, H3K4me3, and CTCF) that are available in the ENCODE Project Consortium. Chromatin functional states are annotated based on the study by *Moudgil et al., 2020*. For this cell line, we assigned states 1–9 and 15 to be in the A (active) because they are related to gene transcription. States 10–14 are repressed, and hence may be classified as repressive (B) (*Appendix 5—figure 8*). Locations of the chromatin loops for HCT 116 from Hi-C contact maps were determined using HiCCUPS in juicer (*Durand et al., 2016*;; https://github.com/aidenlab/juicer) following the procedure from *Rao et al., 2017*. Using this procedure, we detected 3624 loop anchors for 23 chromosomes from WT Hi-C contact maps.

### Calculation of the 3D structures of chromosome from Hi-C contact maps

In order to identify the A/B clusters and calculate the boundary probability, we need the 3D coordinates of the loci. We used the HIPPS (*Shi and Thirumalai, 2021*) method, which uses the Hi-C contact map as the input, to generate an ensemble of 3D chromosome structures. In the HIPPS method, the mean distance matrix is generated using polymer physics using a power-law relation between the mean contact probability, $\langle p_{ij} \rangle$, between loci $i$ and $j$, and the average spatial distance, $\langle r_{ij} \rangle$. An ensemble of 3D structures is calculated by using the mean distance matrix as a constraint using the principle of maximum entropy. We generated an ensemble of 10,000 individual 3D structures by applying the HIPPS method to Hi-C contact map for each chromosome for the two cell lines. The 3D structures are used to calculate the boundary positions (see below) to identify potential single-cell domains, which complements the TopDom analysis.

## Appendix 2

### Effect of *RAD21* removal

Single-cell studies (*Bintu et al., 2018*; *Xie et al., 2020*; *Conte et al., 2020*) on a 2.5 Mb region (Chr21: 34.6–37.1 Mb) in HCT-116 cell showed that several pronounced TAD structures detected in the WT cells were eliminated if *RAD21* is degraded. This could be predicted by the EMH because the degraded TADs are mostly composed of active (A) loci. Our analysis for the same region also shows flat domain boundary probability (*Appendix 5—figure 13a*). In contrast, *Appendix 5—figure 13b* reveals that preferential TAD boundaries persist and are retained despite *RAD21* loss, which is associated with epigenetic switches. Furthermore, some TADs without epigenetic switches are preserved and could be identified based on the presence of physical boundaries in the 3D structures (green lines in *Appendix 5—figure 13c*).

## Appendix 3

### Corner dots in P-TADs

Our analyses of the experimental data show that TADs whose borders correspond to both epigenetic switch and CTCF loop anchors in the WT cells are preserved after removal of cohesin. However, not all P-TADs have loop anchors at their boundaries (corner dots) in the WT cells (see **Appendix 5—figure 12**). Out of the 280 (396) P-TADs with epigenetic switches in mouse liver (HCT-116) chromosomes, 117 (170) did not have corner dots at their boundaries before deleting *Nipbl* (*RAD21*). This suggests that some TAD boundaries are formed in the absence of corner dots. It is the underlying epigenetic landscape that is the predominant factor in the formation of such domain boundaries.

## Appendix 4

### Loss of cohesin-associated CTCF loops leads to an increase in the degree of compartmentalization

The first experimental finding is that cohesin knockdown results in the preservation or even enhancement of the compartment structure (*Schwarzer et al., 2017*; *Rao et al., 2017*; *Wutz et al., 2017*; *Zuin et al., 2014*; *Bintu et al., 2018*). To ensure that CCM reproduces this finding, we first explored how cohesin-associated CTCF loop deletion affects compartmentalization by performing simulations without loop anchors, which is implemented by setting $P_L = 0$ (see 'Methods'). We find that the plaid patterns persist after the deletion of the loops (*Appendix 5—figure 4a*). Visual comparison between the two contact maps in *Appendix 5—figure 4a* shows that this is indeed the case. Moreover, the rectangles in *Appendix 5—figure 4a* also show that finer features, absent when $P_L$ is unity, emerge when $P_L = 0$, which is an indication of the increase in the compartment strength. Pearson correlation maps (lower panel in *Appendix 5—figure 4a*) illustrate the reduction in the contact profiles between A and B loci (dark-blue color) after loop loss. There is an enhancement in interactions between same type loci (A–A or B–B) (dark-red color). The results in *Appendix 5—figure 4a* confirm the experimental finding that disruption of the loops not only creates finer features in the compartment structures but also results in increased number of contacts between loci of the same type, which is simultaneously accompanied by a decrease in the number of A–B interactions.

We then investigated whether the enhancement of compartments observed in the contact maps has a structural basis. An imaging study (*Xie et al., 2022*) showed that active loci form larger spatial clusters in cohesin-depleted cells. To probe whether this finding is reproduced in our CCM simulations, we first calculated the spatial distance matrix using 10,000 simulated 3D structures. For individual matrices, with and without cohesin-associated CTCF loops, we identified A (B)-dense regions as A (B) clusters, respectively, using the DBSCAN (method *Sander et al., 1998*). *Appendix 5—figure 4b* shows the number of clusters obtained using DBSCAN. The number of A clusters varied between individual structures, which reflects the heterogeneity in the chromosome organization. On average, there are about five A clusters for $P_L = 1$, which decreases to four with $P_L = 0$, an indication of the increase in the compartment strength. We also calculated the size of the clusters, which is the average fraction of loci in each cluster in individual structures (right panel in *Appendix 5—figure 4c*). On average, the size of A clusters is greater after loop loss. This implies that active clusters merge in space to form larger and more connected clusters upon deleting the loops. In contrast, there is no change in the number and size of B clusters (see *Appendix 5—figure 4a*). Taken together, our observations for A and B clusters show that the A loci form larger clusters upon loop loss, which leads to stronger segregation between A and B loci after the loops are deleted.

In *Appendix 5—figures 4 and 5*, we used the DBSCAN method to demonstrate the enhanced compartmentalization upon cohesin deletion. In order to assess whether our method produces results that are consistent with other techniques, we also calculated the compartment strength used in previous studies (*Belaghzal et al., 2021*; *Abramo et al., 2019*). In this method, chromosomal interaction frequencies are first normalized by the average interaction frequency at a given genomic distance, $s$. Then the distance-corrected interaction frequencies are sorted based on the PC1 value for each locus $i$, which we calculated from the contact map. Finally, the frequencies were aggregated into 50 bins to obtain the saddle plot (*Appendix 5—figure 6*) in which the top-left corner (B–B) indicates the frequency of interaction between the B compartments, while bottom-right (A–A) represents the contact frequency between the A compartments. Top-right and bottom-left corners are interaction frequencies between A and B compartments. The strength of the compartment is calculated using ((AA) + (BB))/((AB) + (BA)). The values used for this ratio were determined by calculating the mean value of 20% bins in each corner of the saddle plot. We used cooltools for saddle plot calculations (https://github.com/open2c/cooltools, *Abdennur et al., 2022*).

### Relation to experiments regarding the compartmentalization

Besides being consistent with imaging experiments (*Xie et al., 2022*), we wondered whether the structural changes inferred from the Hi-C maps from the mouse liver and HCT-116 cell lines liver (*Schwarzer et al., 2017*; *Rao et al., 2017*) could be used to demonstrate the strengthening of compartments upon deleting cohesin. We calculated an ensemble of 10,000 3D structures for chromosomes from the two cell lines using the HIPPS method (*Shi and Thirumalai, 2021*). Consistent with the simulation results, we also observed variations in the spatial organization of A and B loci

in individual structures in WT and cohesin-depleted cells. DBSCAN analysis identified a smaller number A/B clusters with larger size in Δ*Nipbl* (Δ*RAD21*) cells compared to WT cells, as shown in *Appendix 5—figure 5a* for Chr11 and Chr19 (see *Appendix 5—figure 5b* for HCT-116 results).

To provide additional insights into the calculated conformations, we constructed Pearson correlation matrices using contact maps calculated from the 3D structures. *Appendix 5—figure 5c and d* show that cohesin loss induces stronger plaid patterns, which is consistent with simulation results (lower panel of *Appendix 5—figure 4a*) for Chr13 from GM12878 cell line. A comparison of the 3D structural changes with and without cohesin shows that the micro-phase separation between active and inactive loci is strengthened, with larger A and B physical clusters upon depletion of cohesin, which accords well with Hi-C experiments (*Xie et al., 2022*).

## Appendix 5

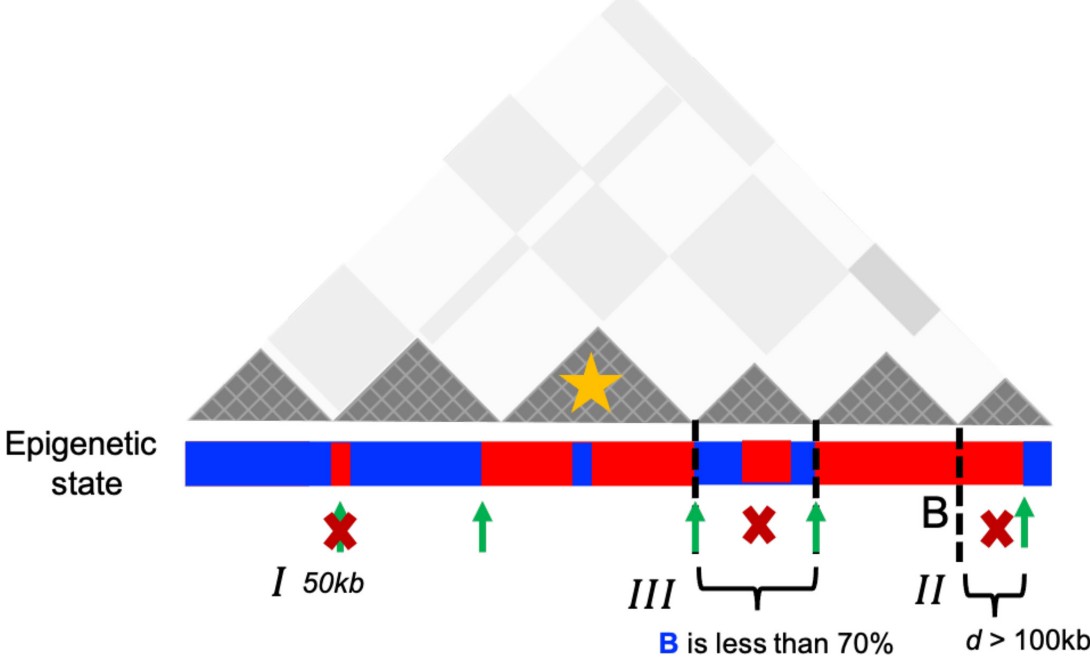

**Appendix 5—figure 1.** Schematic representation used to identify the preserved topologically associating domains (P-TADs) with epigenetic switches. Dark gray triangles represent the P-TADs in contact map. Small square within each triangle represents a single locus (50 kb). Red (blue) color indicates the active (inactive) state in the bar below the contact map. A transition between A and B epigenetic states is referred to as epigenetic switch (green arrows). We examined whether each P-TAD has an epigenetic switch at the boundaries ±100 kb (*II*). If P-TADs have only one locus (50 kb) switch near their boundaries (*I*) or comprise <70% of sequences in identical epigenetic state (*III*), they are excluded. The TAD (yellow star) is a P-TAD with epigenetic switch at the TAD boundary.

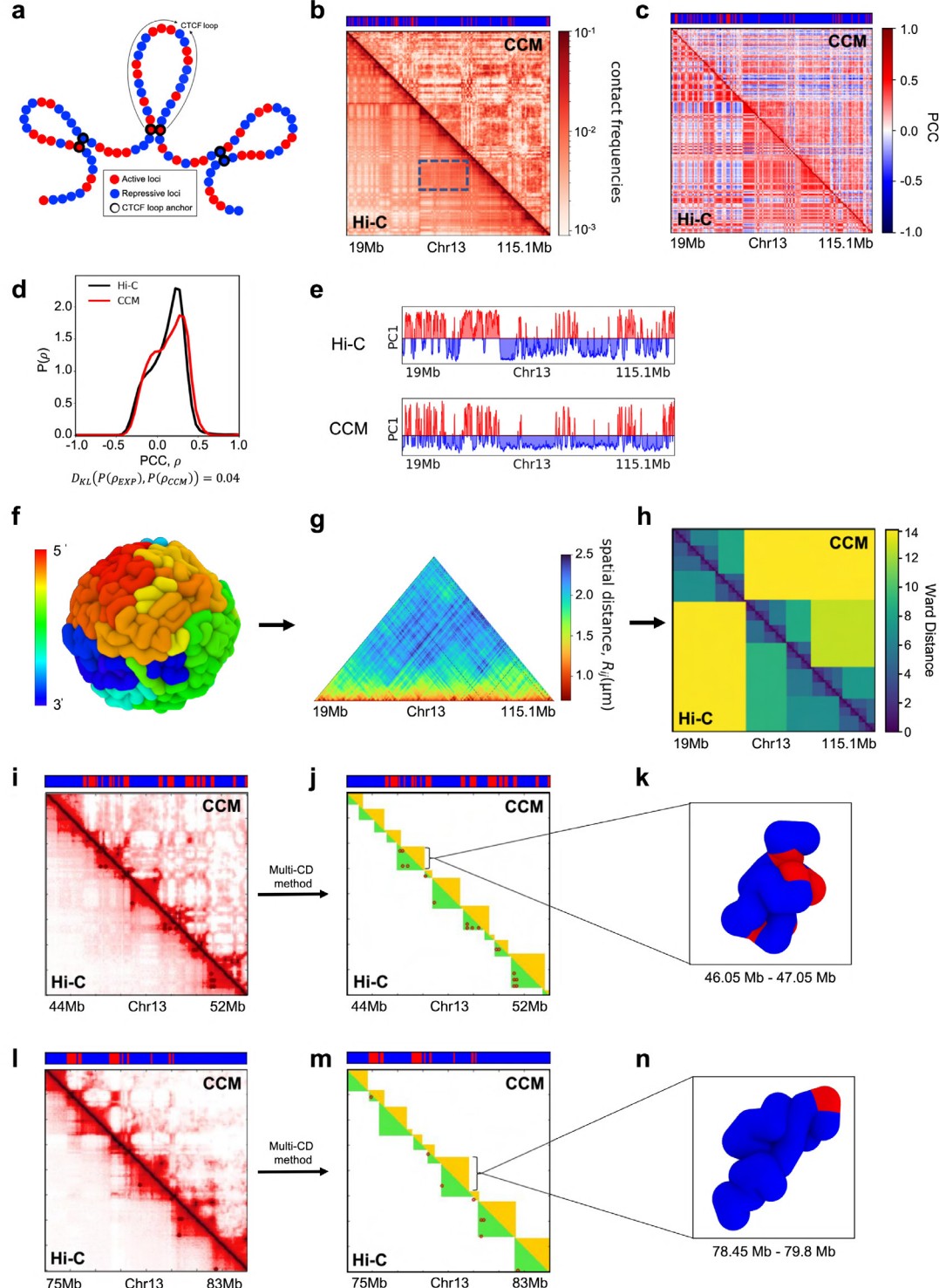

**Appendix 5—figure 2.** Chromosome copolymer model (CCM) simulations for chromosome 13 (Chr13) from the GM12878 cell line. (**a**) In the CCM, red (blue) spheres represent active (repressive) loci. The black open circles are the CTCF loop anchor locations. (**b**) Comparison of the simulated ($P_L = 1$, top half) and Hi-C contact maps (bottom half). The bar above marks the epigenetic states with red (blue) representing active (repressive) loci. The values of the contact frequencies, converted to a $log$ scale, are shown on the right. (**c**) Comparison between the Pearson correlation maps consisting of $\rho_{ij}$ for all loci pairs from simulations (top half) and experimental data (bottom half). The scale for the Pearson correlation coefficient (PCC) is on the right. (**d**) Distribution of the PCC, $\rho_{ij}$, for all $(i, j)$ pairs from simulations and experiment (1 is positive correlation, 0 is no correlation, and −1 corresponds

*Appendix 5—figure 2 continued on next page*

*Appendix 5—figure 2 continued*

to anti-correlation). The Kullback–Leibler, $D_{KL}$, value between CCM prediction and experiment is small. (**e**) First eigenvector values (PC1) from principal component analysis (PCA) using the correlation matrix for CCM. The compartments A and B are defined by positive (red) and negative (blue) values. (**f**) Snapshot of the folded Chr13. The color corresponds to genomic distance from one end point, ranging from red to green to blue. (**g**) Ensemble averaged distance map obtained from simulations. (**h**) Ward linkage matrix (WLM) comparison between simulations and the one computed using Hi-C data. The PCC between the two distance matrices is ~0.83, indicating reasonable agreement between simulations and experiments. (**i**) Contact map for the 8 Mbp region ((44–52) Mb) with the upper (lower) triangle corresponding to simulations (experiments). (**j**) On the right is an Illustration of the TADs, identified using the Multi-CD method (*Bak et al., 2021*). The dark-red circles are the positions of the loop anchors detected in the Hi-C experiment, which are formed by two CTCF motifs. A subset of TADs is defined by the CTCF loops, whereas others are not associated with loops. These could arise from segregation between the chromatin states of the neighboring domains in certain experimental studies (*Rowley et al., 2017*; *Beagan and Phillips-Cremins, 2020*; *Rao et al., 2017*). The average sizes of the TADs detected using Multi-CD method from Hi-C and simulated contact maps are ~750 kbs and ~700 kbs, respectively. (**k**) Snapshot of the TAD, marked in (**j**). (**m**) Same as (**j**) except the TADs were calculated for the region ((75–83) Mb) in (**l**). (**n**) Snapshot of the TAD, marked in (**n**).

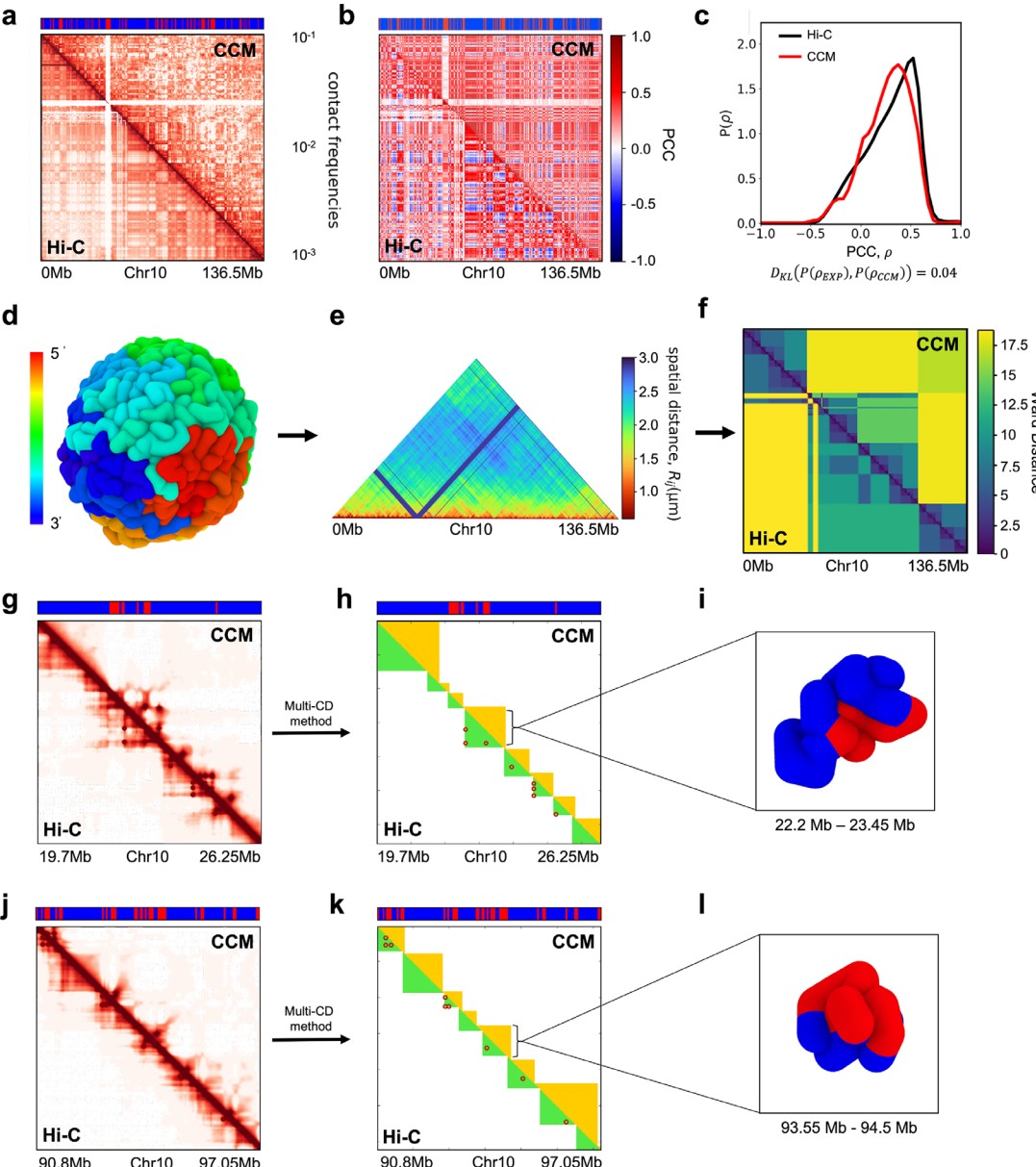

**Appendix 5—figure 3.** Organizational features of Chr10 from human cell line GM12878. (**a**) Comparison between the simulated contact map ($P_L = 1.0$, top half) and Hi-C experiments (bottom half). The bar above the contact map shows the epigenetic states with red (blue) representing active (repressive) loci. (**b**) Experimental (lower triangle) and the simulated (upper triangle) Pearson correlation maps. (**c**) The distribution of the Pearson correlation coefficient (PCC), $\rho_{ij}$ for each pair of $(i, j)$ from simulations and experiment. The value of the Kullback–Leibler (KL) divergence at the bottom is obtained by comparing the distributions obtained in the simulations and experiments. (**d**) A conformation of the folded Chr10 (N = 2712) obtained using the chromosome copolymer model (CCM) simulations. The colors correspond to genomic distance from the 5′ to 3′ end. (**e**) Ensemble averaged distance map calculated using the simulated structures. (**f**) Experimental (lower triangle) and simulated (upper triangle) Ward linkage matrices (WLMs). The PCC between the two WLMs is ~0.75. The agreement between simulations and experiments is fair. (**g**) Hi-C map for the region (19.7–26.25) Mb, with the upper (lower) triangle corresponding to simulations (experiments). (**h**) Right is an illustration of the topologically associating domains (TADs). The dark-red circles are the positions of the loop anchors detected in the Hi-C experiment, formed by two CTCF motifs. (**i**) Snapshot of the TAD, marked by the black line in (**h**). (**k**) Same as (**h**) except the TADs were calculated for a region (90.8–97.05) Mb in (**j**). (**l**) Snapshot of the TAD, marked by the black line in (**k**). The diversity of TAD structures is apparent.

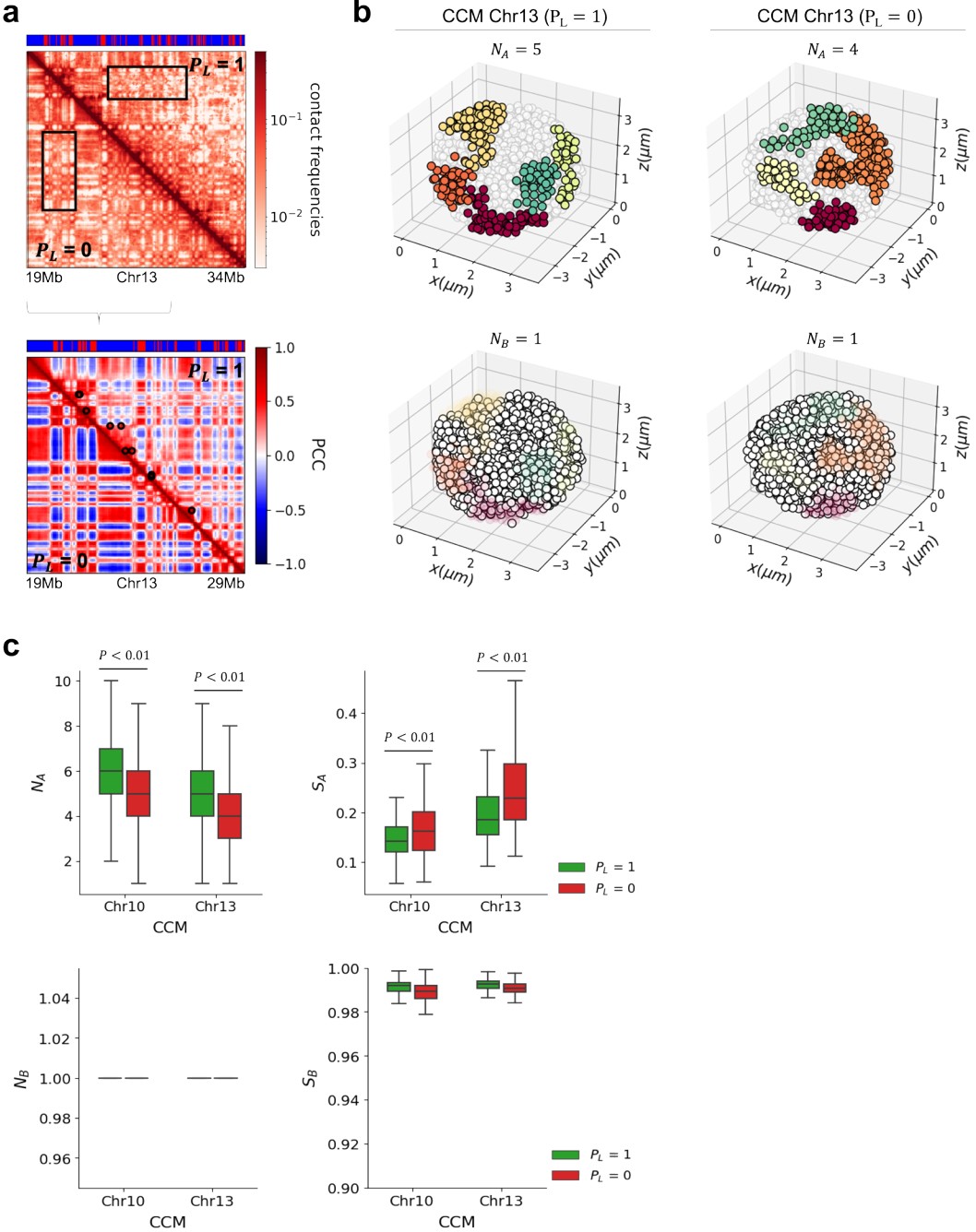

**Appendix 5—figure 4.** Clustering of A and B loci is stronger after loop (cohesin) loss. (**a**) Comparison between simulated contact maps using chromosome copolymer model (CCM) (19–34 Mb, upper panel) and Pearson correlation maps (19–29 Mb, lower panel) for Chr13 (GM12878 cell line). Upper triangle (lower triangle) was calculated with (without) CTCF loops. The black circles in the upper triangle are the positions of the CTCF loop anchors detected in the Hi-C experiment (*Rao et al., 2014*). The bar on top marks the epigenetic states with red (blue) representing active (repressive) loci. Upon CTCF loop loss, the plaid patterns are more prominent, and finer details of the compartment organization emerge. (**b**) 3D snapshots of A and B clusters identified using the density-based spatial clustering of applications with noise (DBSCAN) algorithm, with $P_L = 1$ (left panel) and $P_L = 0$ (right panel) computed from simulations of Chr13 with and without loops, respectively. Five A clusters (upper panel; red, orange, yellow, dark-green, light-green) and one B cluster (lower panel; white) were detected in this 3D structure with $P_L = 1$. Four A clusters and one B cluster were detected for $P_L = 0$. The size of a locus $\sigma_{50K} \approx 243$ nm (*Shi and Thirumalai, 2021*). (**c**) Box plot of the number (left) and average size (right) of A (B) clusters determined

*Appendix 5—figure 4 continued*

using 10,000 individual 3D structures for $P_L = 1$ and $P_L = 0$ for simulated Chr10 and Chr13. The size of the A (B) cluster, $S_A$ ($S_B$), is defined as (the number of A (B) loci within the cluster)/(the total number of A (B) loci within the chromosome). Boxes depict median and quartiles. The black line with caps describes the range of values in the number and size. Loop loss creates a smaller number (enhancement in compartment strength) of A-type clusters whose sizes are larger (upper). Two-sided Mann–Whitney *U* test was performed for the statistical analysis. There is no change in the number and size of B clusters after loop deletion (lower).

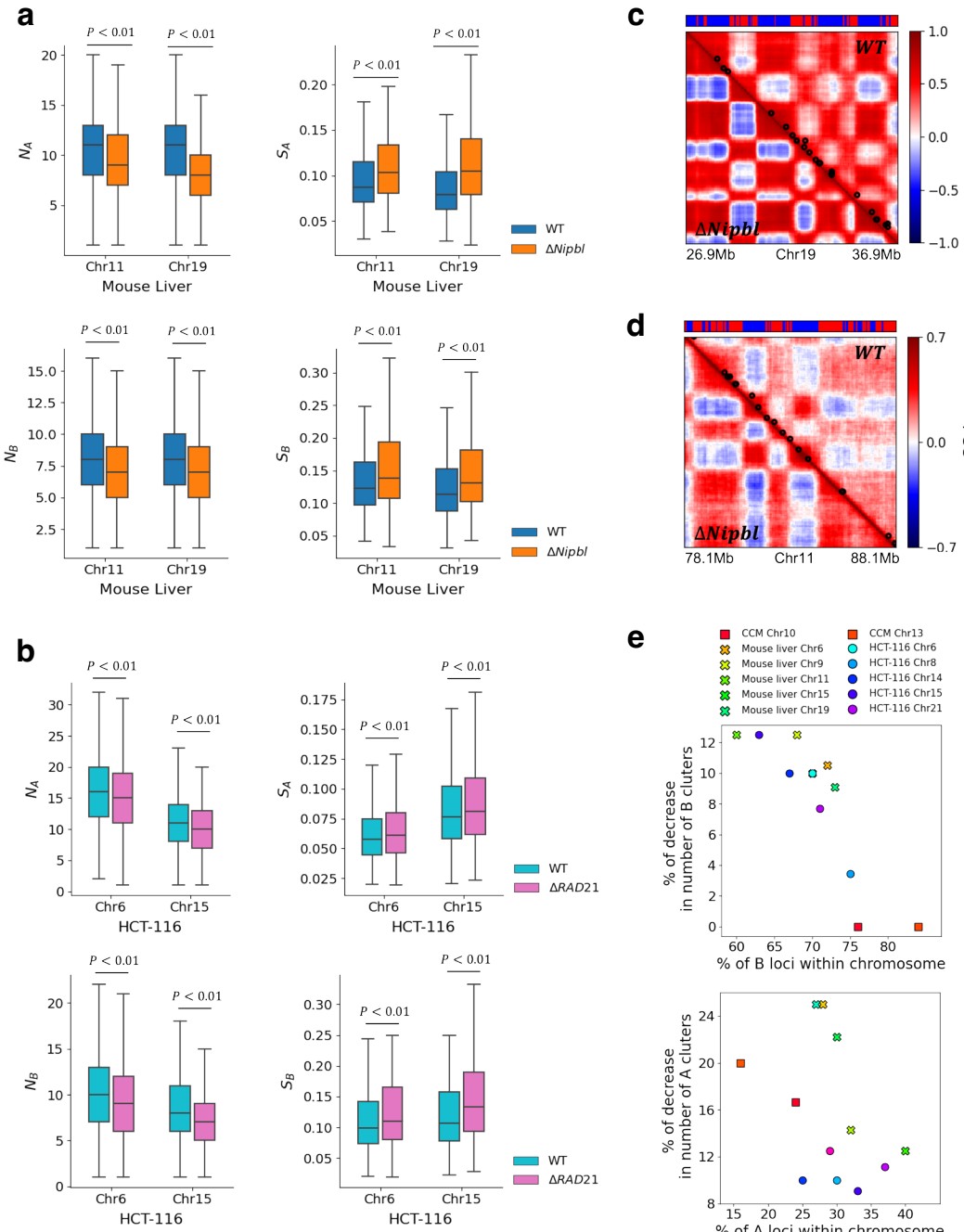

**Appendix 5—figure 5.** Clustering of A and B loci is stronger after loop (cohesin) loss. (**a, b**) Same as *Appendix 5—figure 4c* except the results were determined using 10,000 3D structures generated with the Hi-C-polymer-physics-structures (HIPPS) method from the experimental Chr11 and Chr19 contact maps (Chr6 and Chr15 *Appendix 5—figure 5 continued on next page*

*Appendix 5—figure 5 continued*

contact maps) from mouse liver for the wild-type (WT) and Δ*Nipbl* (**Schwarzer et al., 2017**) (HCT-116 in WT and Δ*RAD21* cells [**Rao et al., 2017**]), respectively. The number of A clusters decreases by 18 and 27% after *Nipbl* loss in Chr11 and Chr19, respectively. (**c, d**) Pearson correlation matrix derived from 3D structures for Chr11 and Chr19 of mouse liver, respectively. Two loci, separated by a distance smaller than 1.75σ, are in contact (σ is the mean distance between $i$ and $i+1$ loci for WT and Δ*Nipbl*, respectively). The black circles in the upper triangle are loop anchors detected in Hi-C map (**Schwarzer et al., 2017**) using HiCCUPS (**Rao et al., 2014**). (**e**) The percentage of decrease in the number of A (B) clusters after CTCF loop or cohesin loss for some chromosomes in simulations and experiments as a function of the percentage of A (B) loci within the chromosome. When the proportion of B loci is much larger than A loci, there is no change in B clusters despite loop or cohesin deletion (upper panel).

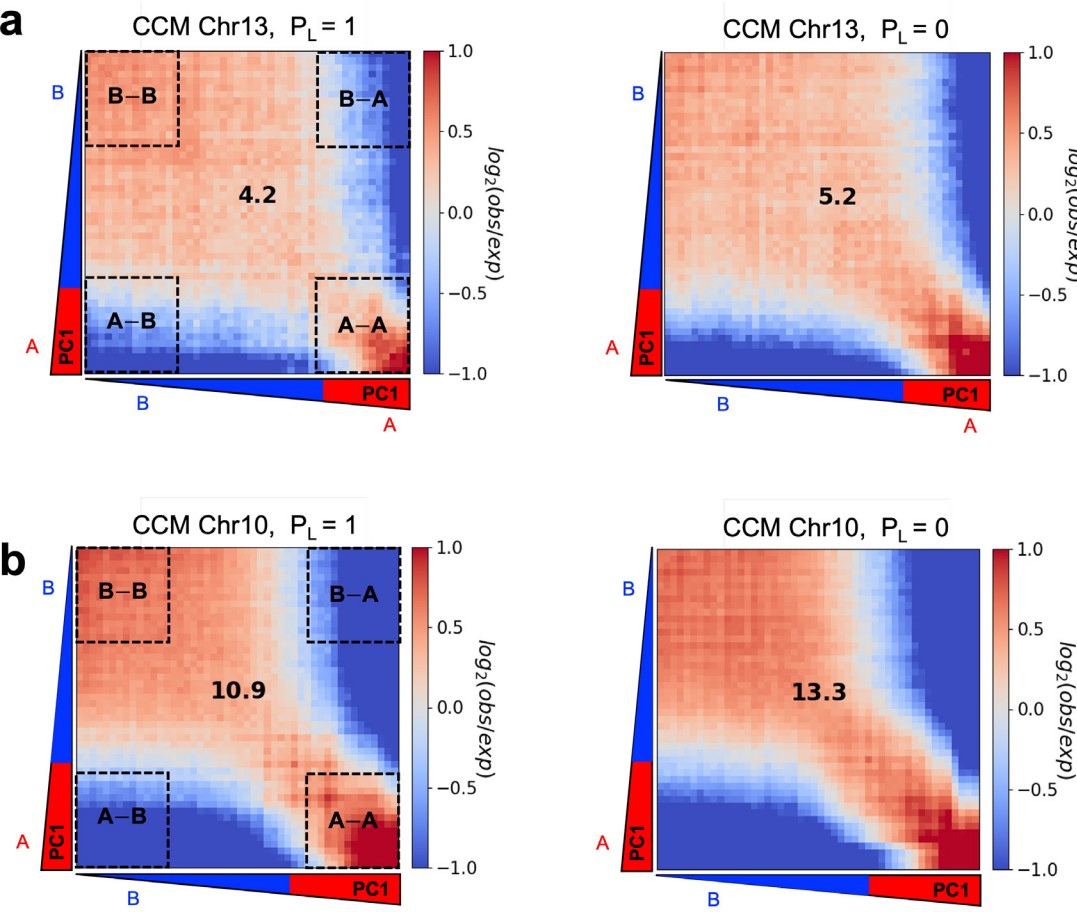

**Appendix 5—figure 6.** Enhancement of compartmentalization upon CTCF loop loss. Compartmentalization saddle plots are shown for (**a**) Chr13 and (**b**) Chr10 with $P_L = 1$ (left) and $P_L = 0$ (right). Observed/expected matrix bins are arranged based on PC1, obtained from the contact maps without loops. Numbers at the center of the maps represent compartment strengths defined as the ratio of ((AA) and (BB) interactions) to ((AB) and (BA) interactions) using the mean values from the corners. The increase in the compartment score (4.2–5.2 for Chr13 and 10.9–13.3 for Chr10) shows that the compartment features are accentuated in $P_L = 0$ (loop deletion) compared to $P_L = 1$, which accords well with the conclusions in the main text that uses a different method.

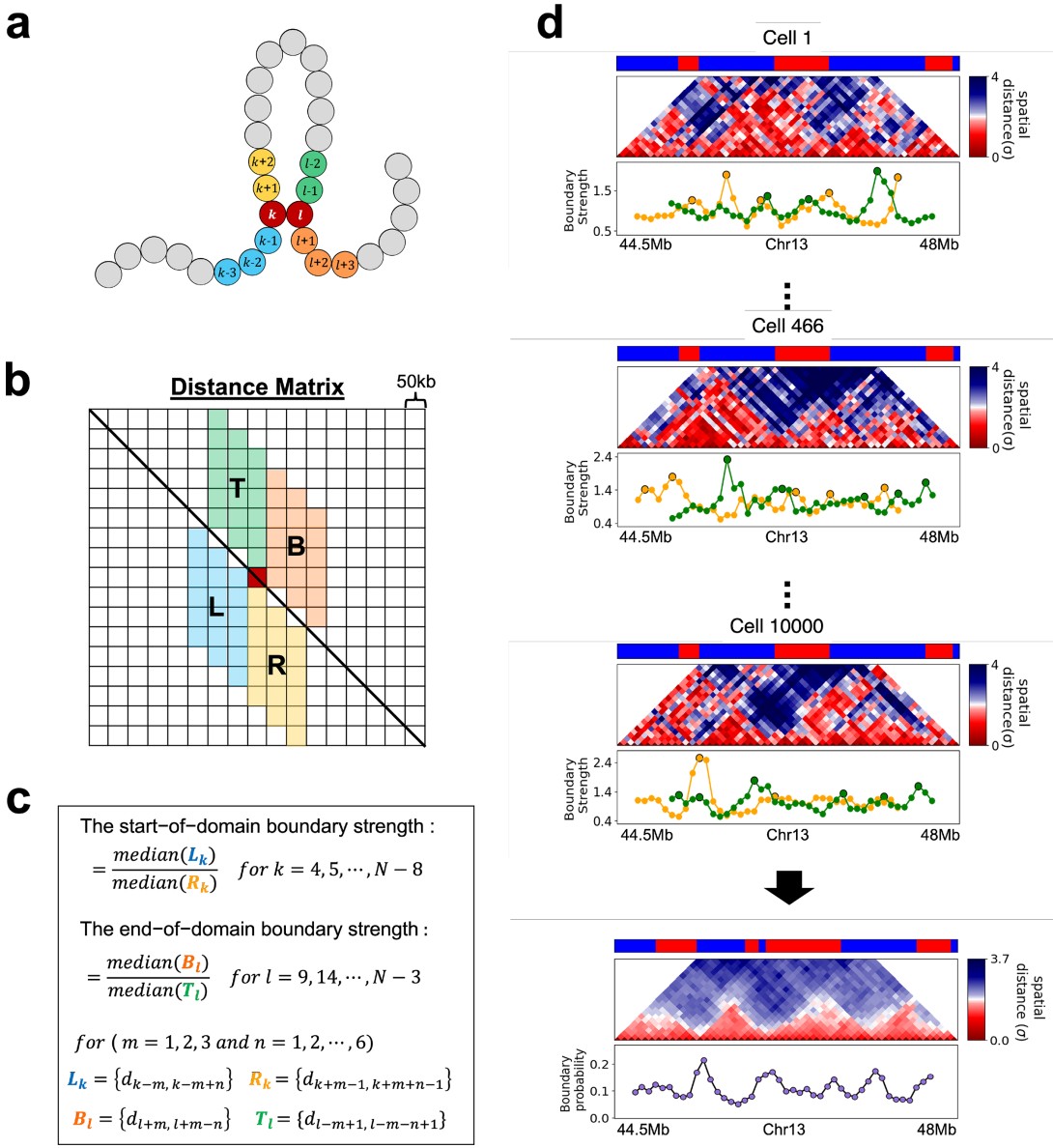

**Appendix 5—figure 7.** Calculation of boundary strength and boundary probability from the distance matrix at 50 kb resolution. (**a**) A schematic describing the chromosome model. (**b**) Each small square of size $a$ (=50 kb) represents distance, $r_{ij}$ between two loci $i$ and $j$. The red square is used to illustrate the idea. (**c**) Definition of the start and end of domain boundary strengths in the $N \times N$ distance matrix. The distance between the loci is represented as arcs in various colors. (**d**) The distance maps in 10,000 cells are calculated using the 3D structures using the HIPPS method (**Shi and Thirumalai, 2021**) with Hi-C contact map from **Schwarzer et al., 2017** as input. Local maxima above a defined threshold at the start/end of domain boundary strengths (yellow and green lines, respectively) are defined as domain boundaries in the WT Chr13. The start/end boundary probabilities for each locus are calculated as the proportion of cells in which the corresponding locus is a boundary location. The average of the start and end boundary probabilities covers 10,000 cells and is defined as the boundary probability for a given locus.

| State | CTCF | H3K9me2 | H3K9me3 | H3K27me3 | H3K36me3 | H4K20me1 | H3K4me1 | H3K4me2 | H3K4me3 | H3K27ac | H3K9ac | H3K79me2 | |
|---|---|---|---|---|---|---|---|---|---|---|---|---|---|
| 1 | 4 | 1 | 1 | 0 | 11 | 2 | 19 | 15 | 1 | 46 | 3 | 7 | Transcribed |
| 2 | 6 | 1 | 2 | 1 | 4 | 2 | 91 | 99 | 45 | 96 | 51 | 5 | Enhancer |
| 3 | 5 | 0 | 1 | 0 | 3 | 2 | 29 | 100 | 100 | 94 | 97 | 42 | Promoter |
| 4 | 17 | 1 | 1 | 3 | 1 | 1 | 38 | 98 | 92 | 22 | 32 | 3 | Promoter |
| 5 | 2 | 3 | 2 | 2 | 2 | 2 | 56 | 58 | 2 | 8 | 2 | 2 | Enhancer |
| 6 | 2 | 3 | 5 | 2 | 23 | 13 | 75 | 84 | 26 | 56 | 23 | 93 | Enhancer |
| 7 | 1 | 3 | 4 | 1 | 14 | 11 | 10 | 1 | 0 | 8 | 1 | 77 | Transcribed |
| 8 | 0 | 1 | 1 | 0 | 3 | 2 | 2 | 0 | 0 | 1 | 0 | 5 | Transcribed |
| 9 | 1 | 4 | 8 | 2 | 50 | 12 | 7 | 0 | 1 | 10 | 0 | 19 | Transcribed |
| 10 | 0 | 17 | 15 | 8 | 6 | 11 | 11 | 1 | 2 | 2 | 0 | 6 | inactive |
| 11 | 0 | 2 | 2 | 1 | 0 | 1 | 2 | 0 | 0 | 0 | 0 | 0 | inactive |
| 12 | 0 | 0 | 0 | 0 | 0 | 0 | 0 | 0 | 0 | 0 | 0 | 0 | inactive |
| 13 | 0 | 1 | 18 | 1 | 2 | 4 | 3 | 0 | 1 | 0 | 0 | 1 | Repressed |
| 14 | 0 | 4 | 3 | 34 | 1 | 5 | 2 | 0 | 0 | 0 | 0 | 1 | Repressed |
| 15 | 89 | 1 | 2 | 2 | 1 | 2 | 12 | 18 | 1 | 4 | 0 | 1 | Insulator |

Chromatin state (vertical axis label) · Chromatin mark observation frequency (%)

**Appendix 5—figure 8.** ChromHMM chromatin state annotation in HCT-116 cells.

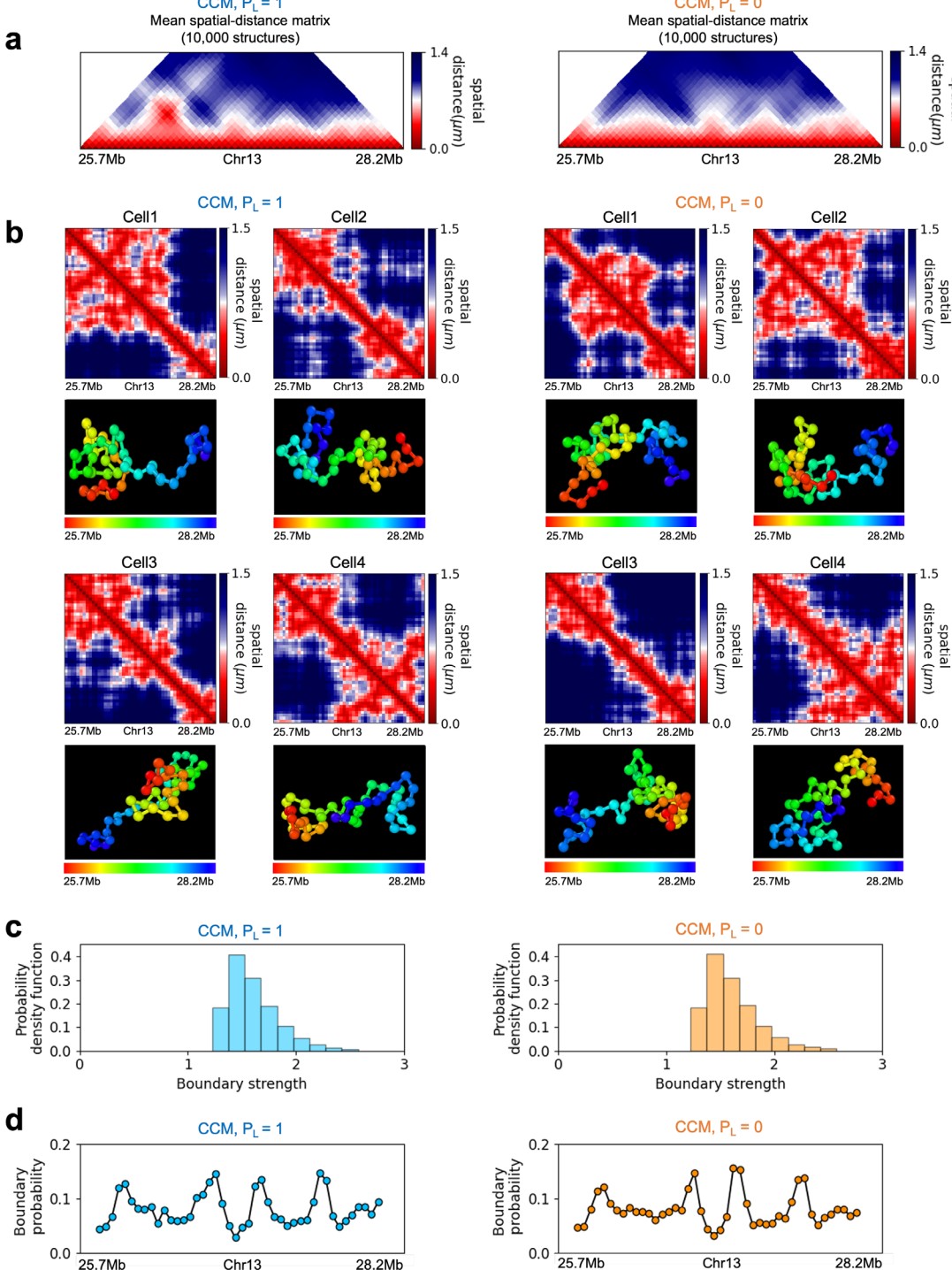

**Appendix 5—figure 9.** Single-cell topologically associating domain (TAD)-like structures are exhibited in both $P_L = 1$ and $P_L = 0$. (**a**) Mean spatial distance matrix for the genomic region (25.7–28.2 Mbps) in chromosome copolymer model (CCM) Chr13 without (left) and with (right) CTCF loops. (**b**) Examples of single-cell spatial distance matrices calculated from the simulated 3D structures. TAD-like structures vary from cell to cell in both $P_L = 1$ (left) and $P_L = 0$ (right). Schematic of structures for the four cells under the two conditions is given below. (**c**) Distribution of the boundary strengths before (left) and after (right) CTCF loop loss, describing the steepness in the changes in the spatial distance across the boundaries. (**d**) The probability for each locus to be a single-cell domain boundary in cells for $P_L = 1$ (left) and $P_L = 0$ (right).

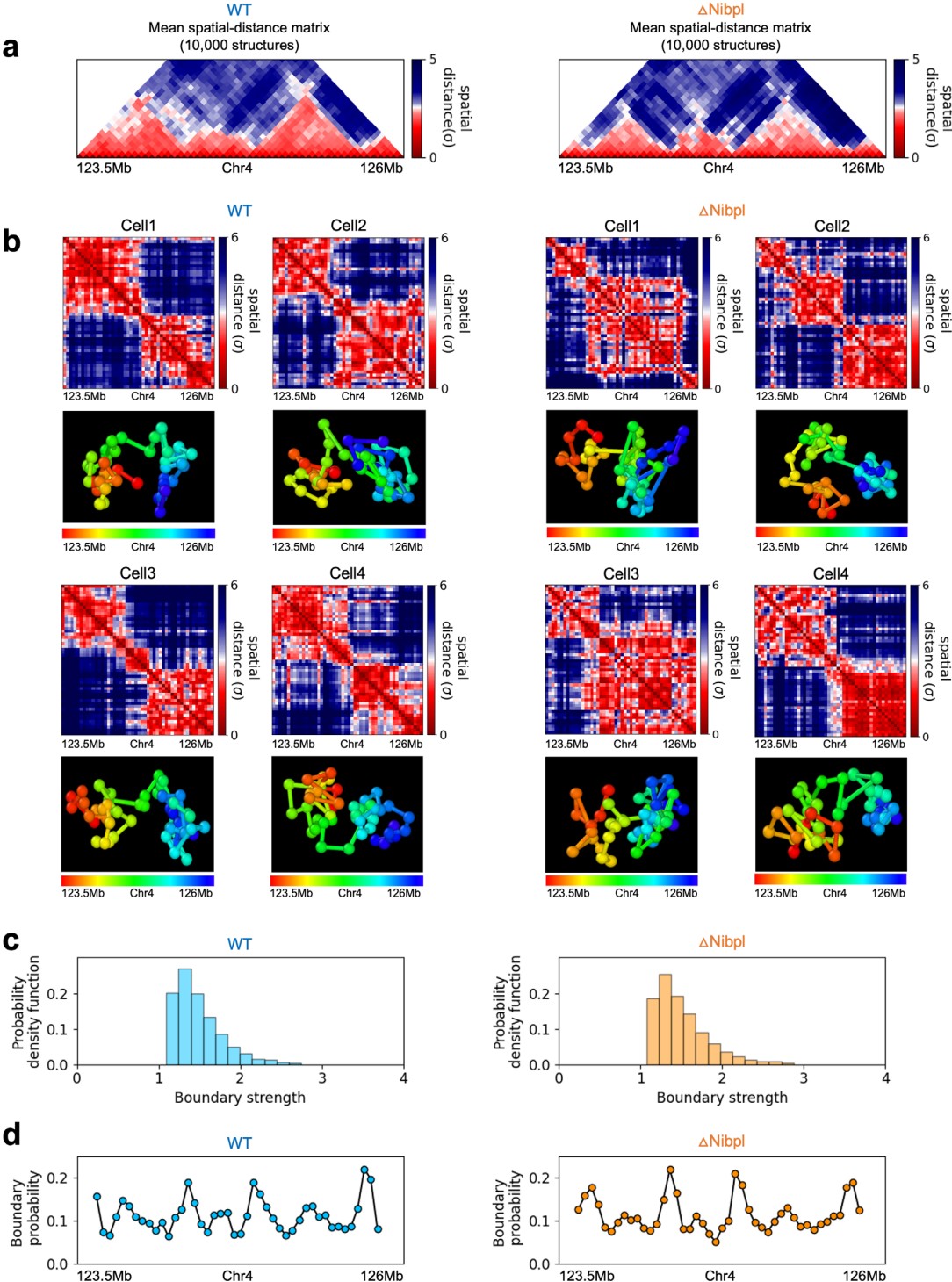

**Appendix 5—figure 10.** Same as *Appendix 5—figure 9* except the results are for the genomic region (123.5–126 Mb) in Chr4 of mouse liver (*Schwarzer et al., 2017*) with (left) and without (right) cohesin loading factor *Nipbl*. Hi-C-polymer-physics-structures (HIPPS)-generated single-cell spatial distance matrices using Hi-C contact maps as inputs.

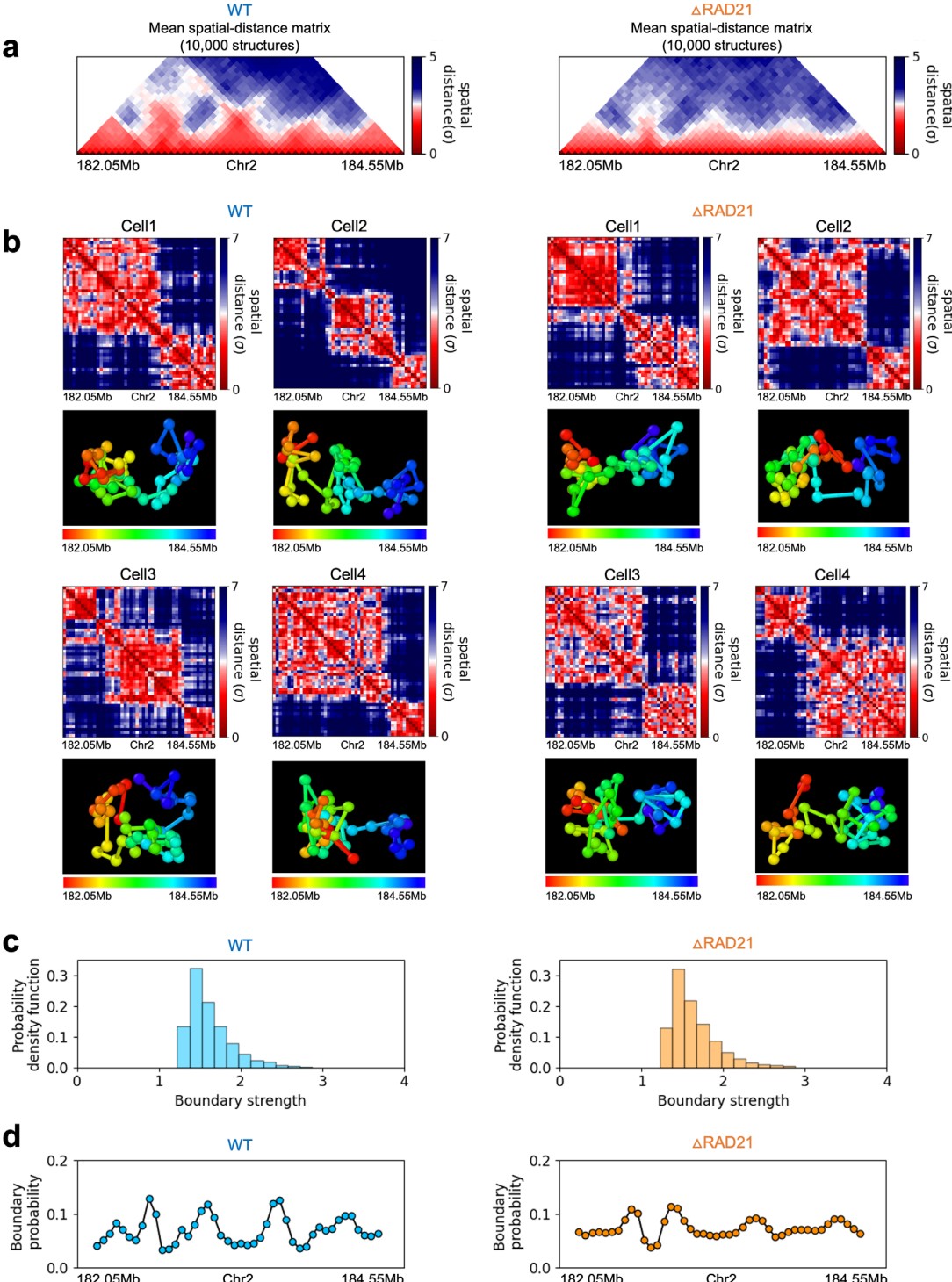

**Appendix 5—figure 11.** Same as *Appendix 5—figure 9* except the results are for the genomic region (182.05–184.55 Mb) in Chr2 of HCT-116 (*Rao et al., 2017*) with (left) and without (right) a core component of the cohesin complex, *RAD21*. Single-cell 3D structures were calculated from Hi-C contact maps using Hi-C-polymer-physics-structures (HIPPS).

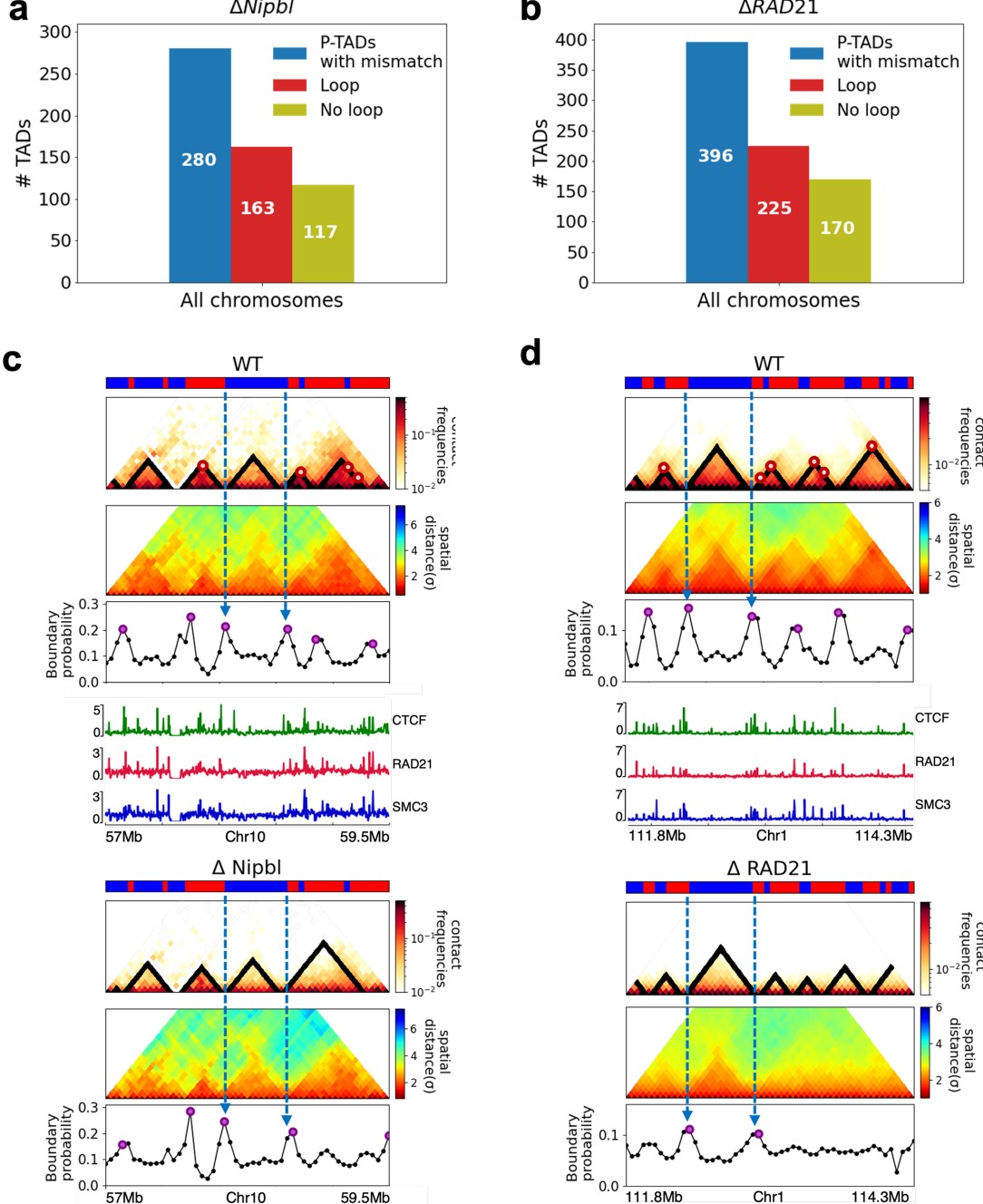

**Appendix 5—figure 12.** Epigenetic states contribute to the formation of domain boundaries. Preserved topologically associating domain (P-TAD) does not always have corner dots at their boundaries in the wild-type (WT) cells. (**a**) The number of P-TADs (after Δ*Nipbl*) whose boundaries coincide with both epigenetic switches and corner dots (CTCF loop anchors) (red color) and only epigenetic switches (olive color) in the WT chromosomes from mouse liver. (**b**) Same as (**a**) except the results are obtained using experimental data from HCT-116 cell. (**c**) Chr10: 57–59.5 Mb in mouse liver and (**d**) Chr1: 111.8–114.3 Mb in HCT-116 cells, respectively. Comparison between 50-kb-resolution contact maps for the 2.5 Mb region with (upper) and without (lower) *Nipbl* (*RAD21*). The panels below show the mean distance maps obtained from the 3D structures. ChIP-seq tracks for CTCF, RAD21, and SMC1 in WT cells (***Schwarzer et al., 2017***; ***Rao et al., 2014***) illustrate the correspondence between the locations of the detected loop anchors and the ChIP-seq signals. Comparison of the contact maps and boundary probabilities in (**c**) and (**d**) shows that the P-TAD boundaries (blue dotted lines) correspond well with epigenetic switch (blue line) even without corner dots in WT cells. Purple circles in the boundary probability graph represent the preferred boundaries.

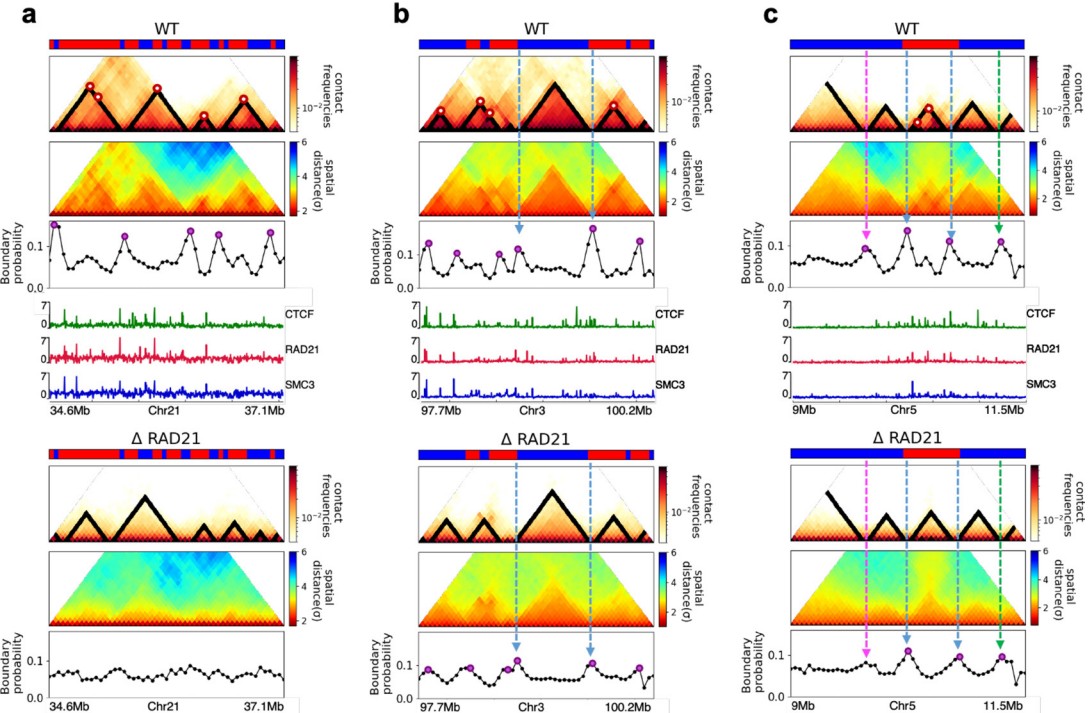

**Appendix 5—figure 13.** Fate of topologically associating domain (TAD) structures after loss of *RAD21* in HCT-116 cells. (**a**) Complete loss (Chr21: 34.6–37.1 Mb). (**b, c**) Preserved (Chr3: 97.7–100.2 Mb and Chr5: 9–11.5 Mb). 50-kb-resolution contact maps for the 2.5 Mb genomic regions of interest with (upper) and without (lower) *RAD21* are shown in the middle panels. The dark-red circles at the boundaries of the TADs in the contact maps are loop anchors detected using HiCCUPS (*Durand et al., 2016*). The mean distance maps calculated using the 3D structures with and without *RAD21* are compared in the top and bottom panels. ChIP-seq tracks for CTCF, RAD21, and SMC1 in WT cells (*Rao et al., 2014*) illustrate the correspondence between the locations of the detected loop anchors and the ChIP-seq signals. Bottom plots are the probability for each genomic position to be a single-cell domain boundary in the regions for cells. Purple circles in the boundary probability graph represent the preferred boundaries. Some P-TAD boundaries match epigenetic switch (blue lines). P-TADs have only high peaks in boundary probability (green line) without evidence for epigenetic switch. The magenta line shows discordance between TopDom and boundary probability.

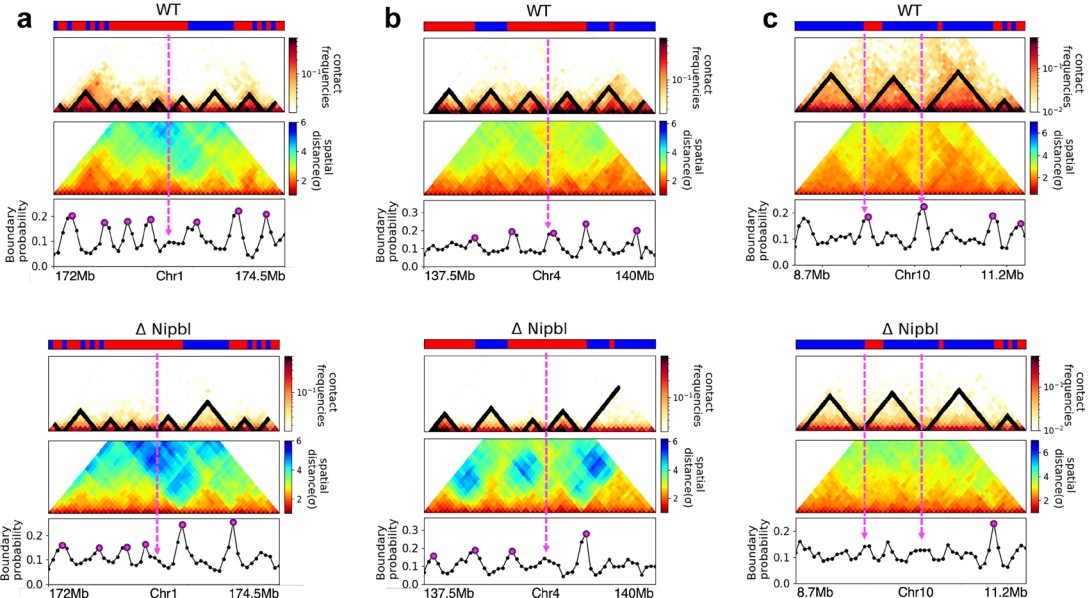

**Appendix 5—figure 14.** Examples of discordance between TopDom and boundary probability predictions in mouse liver (*Schwarzer et al., 2017*). In all cases, the plots show contact maps with TopDom results, mean spatial distance matrix, and boundary probability for the 2.5 Mb region (**a**) (Chr1: 172–174.5 Mb), (**b**) (Chr4: 137.5–140 Mb), and (**c**) (Chr10: 8.7–11.2 Mb) with (top) and without (bottom) *Nipbl*. Purple circles in the boundary probability indicate the prominent physical boundary in 3D structures. The magenta lines represent discordance between TopDom and boundary probability.

