## [Editor Report · eLife assessment]

This **valuable** study, of interest for students of the biology of genomes, uses simulations in combination with published data to examine how many TADs remain after cohesin depletion. The authors suggest that a significant subset of chromosome conformations do not require cohesin, and that knowledge of specific epigenetic states can be used to identify regions of the genome that still interact in the absence of cohesin. The theoretical approaches and quantitative analysis are state-of-the-art, and the data quality and strength of the conclusions are **convincing**, but it is unfortunately still unclear whether physical boundaries (of domains?) in the model appear to be a consequence of preserved TADs, or whether preserved TADs are caused by the physical boundaries.

---

## [Referee Report · Reviewer #1 (Public Review)]

The revised manuscript by Jeong et al presents a thorough analysis of the prevalence and epigenetic causes of TAD conservation upon cohesin loss. The authors suggest that TAD preservation could be caused by an epigenetic switch at the TAD boundary, or by enhancer-promoter or promoter-promoter interactions between TAD boundaries. Simulations using the CCM model confirm that epigenetic switching can mechanistically explain TAD boundary preservation. The added analysis of the prevalence of enhancer and promoter interactions at TAD boundaries strengthens the authors' claim that these interactions could play an important role in TAD preservation.

---

## [Referee Report · Reviewer #2 (Public Review)]

Summary:

Here Jeong et al., use a combination of theoretical and experimental approaches to define molecular contexts that support specific chromatin conformations. They seek to define features that are associated with TADs that are retained after cohesin depletion (the authors refer to these TADs as P-TADs). They were motivated by differences between single cell data, which suggest that some TADs can be maintained in the absence of cohesin, whereas ensemble HiC data suggest complete loss of TADs. By reananalyzing a number of HiC datasets from different cell types, the authors observe that in ensemble methods, a significant subset of TADs are retained. They observe that P-TADs are associated with mismatches in epigenetic state across TAD boundaries. They further observe that "physical boundaries" are associated with P-TAD maintenance. Their structure/simulation based approach appears to be a powerful means to generate 3D structures from ensemble HiC data, and provide chromosome conformations that mimic the data from single-cell based experiments. Their results also challenge current dogma in the field about epigenetic state being more related to compartment formation rather than TAD boundaries. Their analysis is particularly important because limited amounts of imaging data are presently available for defining chromosome structure at the single-molecule level, however, vast amounts of HiC and ChIP-seq data are available. By using HiC data to generate high quality simulated structural data, they overcome this limitation. Overall, this manuscript is important for understanding chromosome organization, particularly for contacts that do not require cohesin for their maintenance, and for understanding how different levels of chromosome organization may be interconnected. I cannot comment on the validity of the provided simulation methods and hope that another reviewer is qualified to do this.

Specific comments

-It is unclear what defines a physical barrier. From reading the text and the methods, it is not entirely clear to me how the authors have designated sites of physical barriers. It may help to define this on pg 7, second to last paragraph, when the authors first describe instances of P-TAD maintenance in the absence of epigenetic mismatch.

-Figure 7 adds an interesting take to their approach. Here the authors use microC data to analyze promoter-enhancer/promoter-promoter contacts. These data are included as part of the discussion. I think this data could be incorporated into the main text, particularly because it provides a biological context where P-TADs would have a rather critical role.

-Figure 3a- the numbers here do not match the text (page 6, second to last paragraph). The numbers have been flipped for either chromosome 10 or chromosome 13 in the text or the figures.

In the revision, the authors have sufficiently addressed my specific concerns from above.

---

## [Referee Report · Reviewer #3 (Public Review)]

This manuscript presents a comprehensive investigation into the mechanisms that explain the presence of TADs (P-TADs) in cells where cohesin has been removed. In particular, to study TADs in wildtype and cohesin depleted cells, the authors use a combination of polymer simulations to predict whole chromosome structures de novo and from Hi-C data. Interestingly, they find that those TADs that survive cohesin removal contain a switch in epigenetic marks (from compartment A to B or B to A) at the boundary. Additionally, they find that the P-TADs are needed to retain enhancer-promoter and promoter-promoter interactions.

Overall, the study is well-executed, and the evidence found provides interesting insights into genome folding and interpretations of conflicting results on the role of cohesin on TAD formation.

---

## [Author Response]

The following is the authors’ response to the original reviews.

**eLife assessment:**
This valuable study, of interest for students of the biology of genomes, uses simulations in combination with published data to examine how many TADs remain after cohesin depletion. The authors suggest that a significant subset of chromosome conformations do not require cohesin, and that knowledge of specific epigenetic states can be used to identify regions of the genome that still interact in the absence of cohesin. The theoretical approaches and quantitative analysis are state-of-the-art, and the data quality and strength of the conclusions are solid. However, because "physical boundaries (of domains?)" in the model appear to be a consequence of preserved TADs, rather than the other way around, the functional insights are limited.Summary of the reviewer discussion for the authors:While the simulations are state of the art and the reviewers appreciated that the approaches used here might help to resolve apparent discrepancies between conclusions from single-cell and bulk/ensemble techniques to study chromosome conformation, the work would benefit from clarification of what precisely is meant with "physical boundaries" and from a comparison of CCM and HIPPS models to understand commonalities and differences between them. In addition, more discussion of the relation of the current work to previous studies, such as Schwarzer et al., 2017, and Nuebler et al., 2018, would elevate the work and make the key claims more compelling. Please see also the detailed comments from the expert reviewers.

We thank the editor for the assessment and the reviewers for the incisive comments. We will address these comments one by one. In particular, we attempt to clarify the concept of “physical boundaries” and its relevance in our study. We hope our responses are satisfactory. We believe that our manuscript has benefitted substantially by revising the manuscript following the comments by the reviewers.

Below is our point-by-point response to the comments:

**Reviewer #1 (Public Review):**
Summary:In this paper, Jeong et al. investigate the prevalence and cause of TADs that are preserved in eukaryotic cells after cohesin depletion. The authors perform an extensive analysis of previously published Hi-C data, and find that roughly 15% of TADs are preserved in both mouse liver cells and in HCT-116 cells. They confirm previous findings that epigenetic mismatches across the boundaries of TADs can cause TAD preservation. However, the authors also find that not all preserved TADs can be explained this way. Jeong et al. provide an argument based on polymer simulations that "physical boundaries" in 3D structures provide an additional mechanism that can lead to TAD preservation. However, in its current form, we do not find the argumentation and evidence that leads to this claim to be fully compelling.Strengths:We appreciate the extensive statistical analysis performed by the authors on the extent to which TAD's are preserved; this does seem like a novel and valuable contribution to the field.

We thank the reviewer for a succinct and comprehensive summary of our work and for appreciating value of our work.

Weaknesses:1. As the authors briefly note, the fact that compartmentalization due to epigenetic mismatches can cause TAD-like structures upon cohesin depletion has already been discussed in the literature; see for example Extended Data Figure 8 in (Schwarzer et al., 2017) or the simulation study (Nuebler et al., 2018). We are hence left with the impression that the novelty of this finding is somewhat overstated in this manuscript.

It is unclear to us by studying the results in the Extended Data Figure 8 that the authors have shown that epigenetic mismatches cause TAD-like structures. As far as we can discern, the data, without a quantitative analysis, shows that may be new TAD-like structures that are not in the wild type appear when cohesin is deleted.

The studies by Schwarzer et al 2017 and Nuebler et al 2018 are relevant to our own investigation, which we undertook after scrutinizing the experiments in Schwarzer et al 2017 and the related work by Rao et. al in 2017 on a different cell line. In the summary of the Reviewer discussion, it is suggested we discuss the relation to the experimental study by Schwarzer et al 2017 and the computational work by Nuebler et al 2018.

(1) The results and the corresponding discussion in these two studies are different (may be complimentary) from our results. When referring to the Extended Data Figure 8 Schwarzer and co-authors state in the main text, “The finer compartmentalization explains most of the remaining or new domains and boundaries seen in Nipbl Hi-C maps”. We are not 100% sure what “remaining” means in this context. The Extended Data Fig. 8(a) shows the “common boundaries” is correlated with the eigenvectors of compartmentalization. If this indeed is what the reviewer is referring to, we believe that our study differs from theirs in two important ways: First, Extended Data Fig.8 (a) is a statistical analysis at the “ensemble” level. In our study, we examined the preservation of TADs at both individual and ensemble level with more detailed analysis. Second, in Extended Data Fig. 8(a), the “common boundaries” (incidentally we are uncertain how that was calculated) are compared to the eigenvectors of PCA analysis of the compartments (larger length scales). In contrast, in our study, we examined the correlation between TAD boundaries and the epigenetic profiles. We believe that this is an important distinction. The PCA analysis of compartments and “common boundaries” defined using (presumably) the insulation score are both derived from the Hi-C contact map. Epigenetic profile, on the other hand, is independent of Hi-C data. We believe our contribution, is to build the connection between epigenetic profiles with the preservation of TADs, and link it to 3D structures. For these reasons, we assert that our results are novel, and are not contained (or even implied) in the Schwarzer et al 2017 study.

The simulations in Neubler et al 2018, which were undertaken to rationalize the experimenrs, revealed that compartmentalization of small segments is enhanced after cohesin depletion, while TADs disappear, which support the broad claims that are made in the experiments. They assert that the structures generated are non-equilibrium. They do not address the emergence of preserved nor the observation of TAD-like structures at the single cell level. However, our goal was to elucidate the reasons for of preservation of TADs. By that we mean, the reasons why certain TADs are present in both the wild and cohesin depleted cells? Through a detailed analyses of two cells, polymer simulations, we have provided a structural basis to answer the question. Finally, we have provided a plausible between TAD preservation and maintenance of enhancer-promoter interactions by analyzing the Micro-C data. For all these reasons, we believe that our study is different from the results in the Extended Figure 8 or the simulations described by Neubler.

Let us summarize the new results in our study that are not contained in the studies referred to by this Reviewer. (1) We showed by analyzing the Hi-C data for mouse liver and HCT-16 that a non-negligible fraction of TAPs is preserved, which set in motion our detailed investigation. (2) Then, using polymer simulations on a different cell type, we generated quantitative insights (epigenetic mismatches as well as structural basis) for the preservation of TADs. Although not emphasized, we showed that deletion of cohesin in the GM12878 cells also give rise to P-TADs a prediction that suggests that the observations might be “universal”. (3) Rather than perform, time consuming polymer simulations, we calculated 3D structures directly from Hi-C data for the mouse liver and HCT-16 cells, which provided a structural basis for TAP preservation. (4) The 3D structures also showed how TAD-like features appear at the single cell level, which is in accord with imaging experiments. (5) Finally, we suggest that P-TADs may be linked to the maintenance of enhancer-promoter and promoter-promoter interactions by calculating the 3D structures using the recent Micro-C data.

For the reasons given above, we assert that our results are novel, and bring new perspectives that are not in the aforementioned insightful studies cited by the Reviewer.

2. It is not quite clear what the authors conceptually mean by "physical boundaries" and how this could offer additional insight into preserved TADs. First, the authors use the CCM model to show that TAD boundaries correlate with peaks in the single cell boundary probability distribution of the model. This finding is consistent with previous reports that TAD-like structures are present in single cells, and that specific TAD boundaries only arise as a population average.The finding based on the CCM simulations hence seems to be that preserved TADs also arise as a population average in cohesin-depleted cells, but we do not follow what the term "physical boundaries" refers to in this context. The authors then use the Hi-C data to infer a maximumentropy-based HIPPS model. They find that preserved TADs often have boundaries that correspond to peaks in the single cell boundary probabilities of the inferred model. The authors seem to imply that these peaks in the boundary probability correspond to "physical boundaries" that cause the preservation of TADs. This argument seems circular; the model is based on inferring interaction strengths between monomers, such that the model recreates the input Hi-C map. This means that the ensemble average of the model should have a TAD boundary where one is present in the input Hi-C data. A TAD boundary in the Hi-C data would then seem to imply a peak in the model's single-cell boundary probability. (The authors do display two examples where this is not the case in Fig.3h, but looking at these cases by eye, they do not seem to correspond to strong TAD boundaries.) "Physical boundaries" in the model are hence a consequence of the preserved TADs, rather than the other way around, as the authors seem to suggest. At the very least the boundary probability in the HIPPS model is not an independent statistic from the Hi-C map (on which their model is constrained), so we have concerns about using the physical boundaries idea to understand where some of the preserved TADs come from.

There are many statements in this long comment that require us to unpack separately. First, using both the CCM simulations, and even more importantly using data-driven approach, we established that TAD-like structures are present in single cells with and without cohesin. The latter finding is fully consistent with imaging experiments. We are unaware of other computational efforts, before our work, demonstrating that this is the case. Perhaps, the Reviewer can point to the papers in the literature.

Regarding the statement that our arguments are circular, and lack of clarity of the meaning of physical boundary, please allow us to explain. First, we apologize for the confusion. Let us clarify our approach. We first used CCM to understand the potential origin of substantial fraction of P-TADs in the GM. The simulations, allowed us to generate the plausible mechanisms, for the origin of P-TADs. Because the CCM does reproduce the Hi-C data, we surmised that the general mechanisms inferred from these simulations could be profitably used to analyze the experiments. The simulations also showed that knowledge of 3D structures produces a muchneeded structural basis of P-TADs , and potentially for emergence of new TADs upon cohesin depletion.

Because 3D coordinates are needed to obtain structural insights into the role of cohesin, we use a novel method to obtain them without the need for simulations. In particular, we used the HIPPS method to obtain 3D coordinates with the Hi-C data as the sole input, which allowed us to calculate directly the boundary probabilities. The excellent agreement between the predicted 3D structures and imaging experiments suggests that meaningful information, not available in Hi-C, may be gleaned from the ensemble of calculated 3D structures.

Although “physical boundary”, a notion introduced by Zhuang, is defined in in the method section, it is apparently unclear for which we apologize. Because this is an important technical tool, we have added a summary in the main text in the revision. We did not mean to imply that the physical boundaries cause the preservation of TADs, although we found that maintenance of the enhancer-promoter contacts (see Fig. 8 in the revision) could be one of the potential reasons for the emergence of physical boundaries. We agree with the reviewer that physical boundaries are structural evidence of preserved TADs (not the cause), that is when a TAD is preserved, we can detect it by prominent physical boundary. The purpose and benefit of physical boundary analysis and using HIPPS in general is to obtain three-dimensional structures of chromosomes. Although both CCM simulations and HIPPS use Hi-C contact maps, three-dimensional structures provide additional information that is not present in the Hi-C data.

The arguments that the authors use to justify their claims could be clarified and strengthened.Here are some suggestions:-Explain the concept of "physical boundaries" more clearly in the main text.

As explained above, we have revised the text to clarify the concept and purpose of physical boundaries analysis. See Page 7.

Justify why the boundary probabilities and the physical boundaries concept can be used to offer novel insight into where preserved TADs may come from.

Boundary probabilities and physical boundaries provide previously unavailable 3D structural information on the TADs structures both at the single-cell and population level. This provides a direct structural basis for determining which TADs are preserved. But in order to understand where P-TADs may come from, physical boundaries analysis alone is not sufficient. As we have shown in the analysis of enhancer-promoter contact, using physical boundary analysis from 3D structures, we can conclude that conservation of enhancer-promoter contact could be one of the reasons for the P-TAD.

Explain more clearly what the additional value of using the HIPPS model to study TAD preservation is.

Our goal, as announced in the title is to elucidate the structural basis for the emergence of PTADs. The HIPPS method, which avoids doing simulations (like CCM and other polymer models used in the literature) provides an ensemble of 3D conformations using averaged contact map generated in Hi-C experiments. Even more importantly, HIPPS produce an ensemble of structures, which can be the basis for predicting the outcomes at the single-cell level. The accuracy of the generated structures has been shown in our previous work (Shi and ThirumalaiPRX 2021). In ensemble-averaged Hi-C experiments, TADs appear to be relatively stable. However, imaging experiments (Bintu et. al, 2018) have revealed that TADs are not fixed structures present in every single cell, but instead exhibit variability at the single-cell level. TADlike structures with distinct boundaries are observed in individual cells, and the location of these boundaries varies from cell to cell. However, these TAD-like structures still show a preferential positioning in 3D structures. Interestingly, the preferential positioning often corresponds to TAD boundaries observed in population-averaged Hi-C data. This suggests that while cohesin is involved in establishing the overall organization of TADs, other factors and mechanisms could also contribute to TAD formation at the individual cell level. In this study, we showed some boundaries of P-TADs upon cohesin loss in the Hi-C maps, align with preferential boundaries in individual 3D structures of chromosomes. The makes the finding that a subset of TADs is preserved upon cohesin is robust.

From a technical perspective, the use of HIPPS avoids time-consuming polymer simulations. The HIPPS is rapid and can be used to generate arbitrarily large ensemble of structures, allowing us calculate properties both at the single cell and ensemble level.

In addition, we'd like to offer the following feedback to the authors.1. The discussion of enhancer-promoter loops as a cause of TAD preservation is interesting, but it would be interesting to know fraction of preserved TADs enhancer-promoter loops might explain.

We thank the reviewer for the excellent suggestion. We have done the suggested calculation. The results are shown in a new Figure.8 in the main text. We also moved the results on enhancer-promoter to the main results section from the Discussion section.

2. The last paragraph of the introduction seems to state that only the HIPPS model was used to find single-cell 3D structures and boundary probabilities. However, the main text suggests that the CCM model was also used for these purposes.

We have revised the text to clarify this point on pages 3-4. Also please see the discussion on the utility of HIPPS above.

3. When referring to the boundary probability, it would be useful if the authors always specified whether they refer to the boundary probability before or after cohesin depletion (or loop depletion in the CCM model). Statements such as "This implies that peaks in the boundary probabilities should correspond to P-TADs" are ambiguous; it is unclear if the authors mean that boundary probabilities before cohesin depletion predict that the boundary will be preserved, rather than that preserved TAD boundaries correlate with peaks in the boundary probability after cohesin depletion.

We thank the reviewer for the suggestion. Indeed, it may be confusing. Hence, we have revised the text in numerous places to clarify this point.

4. It would be interesting to analyze all TAD boundaries that are present after cohesin depletion, rather than just those that overlap with TAD boundaries in WT cells. This would give better statistics for answering the question what causes TAD-like structures in cells without cohesin.

We thank the reviewer for this excellent suggestion. First, this would we believe this deviate from the primary goal of this study: what leads to TAD preservation after cohesin deletion? Second, this has to be done very systematically, as we did here for P-TADs, in order draw meaningful conclusions. This is a very useful study for another occasion.

5. The use of a plethora of acronyms (P-TAD, CM, DM, CCM, HLM...) makes the paper difficult to read.

We have revised the text to change CM to “contact map” and “DM” to “distance map”. For PTADs, CCM, and WLM, we would argue that P-TAD is rather a clear and intuitive abbreviation and CCM/WLM refers to specific methods/models and replacing them with full names would make text more difficult to read. We hope the reviewer is okay with us keeping these acronyms.

**Reviewer #2 (Public Review):**
Summary:Here Jeong et al., use a combination of theoretical and experimental approaches to define molecular contexts that support specific chromatin conformations. They seek to define features that are associated with TADs that are retained after cohesin depletion (the authors refer to these TADs as P-TADs). They were motivated by differences between single cell data, which suggest that some TADs can be maintained in the absence of cohesin, whereas ensemble HiC data suggest complete loss of TADs. By reananalyzing a number of HiC datasets from different cell types, the authors observe that in ensemble methods, a significant subset of TADs are retained. They observe that P-TADs are associated with mismatches in epigenetic state across TAD boundaries. They further observe that "physical boundaries" are associated with P-TAD maintenance. Their structure/simulation based approach appears to be a powerful means to generate 3D structures from ensemble HiC data, and provide chromosome conformations that mimic the data from single-cell based experiments. Their results also challenge current dogma in the field about epigenetic state being more related to compartment formation rather than TAD boundaries. Their analysis is particularly important because limited amounts of imaging data are presently available for defining chromosome structure at the single-molecule level, however, vast amounts of HiC and ChIP-seq data are available. By using HiC data to generate high quality simulated structural data, they overcome this limitation. Overall, this manuscript is important for understanding chromosome organization, particularly for contacts that do not require cohesin for their maintenance, and for understanding how different levels of chromosome organization may be interconnected. I cannot comment on the validity of the provided simulation methods and hope that another reviewer is qualified to do this.

We appreciate the reviewer for a comprehensive summary of our work, and we are happy that the reviewer finds our work important, which provides valuable insights to the field.

Specific commentsIt is unclear what defines a physical barrier. From reading the text and the methods, it is not entirely clear to me how the authors have designated sites of physical barriers. It may help to define this on pg 7, second to last paragraph, when the authors first describe instances of PTAD maintenance in the absence of epigenetic mismatch.

We thank the reviewer for the suggestions. The details of physical boundary designation are provided in the appendix data analysis. To make the concept and idea of physical boundary easy to understand, we have revised the text on page 7 in the revised main text.

Figure 7 adds an interesting take to their approach. Here the authors use microC data to analyze promoter-enhancer/promoter-promoter contacts. These data are included as part of the discussion. I think this data could be incorporated into the main text, particularly because it provides a biological context where P-TADs would have a rather critical role.

We thank the reviewers for the suggestion. We also agree that results in Figure 7 provide novel insights on TAD formation and its possible preservation upon perturbation. We have followed the reviewer’s suggestion to move it to an independent section in the main results section as the last subsection.

Figure 3a- the numbers here do not match the text (page 6, second to last paragraph). The numbers have been flipped for either chromosome 10 or chromosome 13 in the text or the figures.

We thank the reviewer for pointing out this error. In the revised main text, it has been corrected.

**Reviewer #3 (Public Review):**
This manuscript presents a comprehensive investigation into the mechanisms that explain the presence of TADs (P-TADs) in cells where cohesin has been removed. In particular, to study TADs in wildtype and cohesin depleted cells, the authors use a combination of polymer simulations to predict whole chromosome structures de novo and from Hi-C data. Interestingly, they find that those TADs that survive cohesin removal contain a switch in epigenetic marks (from compartment A to B or B to A) at the boundary. Additionally, they find that the P-TADs are needed to retain enhancer-promoter and promoter-promoter interactions.Overall, the study is well-executed, and the evidence found provides interesting insights into genome folding and interpretations of conflicting results on the role of cohesin on TAD formation.

We are pleased with the reviewer’s positive assessment of our work.

To strengthen their claims, the authors should compare their de-novo prediction approach to their data-driven predictions at the single cell level.

We thank the reviewer for the very good suggestion. We are assuming that the Reviewer is asking us to compare the CCM simulations with HIPPS generated structures at the single cell level. We have shown, using the GM12878 cell data, that the polymer simulations reproduce the Hi-C contact maps (an average quantity) well (see Appendix Fig. 2 and Fig. 3). In addition, we show in Appendix Fig. 8 the comparison with ensemble averaged distance maps as well as at the single cell level for Chr 13 from the GM12878 cell. There are TAD-like structures at the single cell level just as we find for HCT-116 cell (Fig. 5 in the main text). Thus, the conclusions from de-novo prediction and data-driven predictions are consistent. In addition, in our previous publication introducing HIPPS in Phys Rev X 11: 011051 (2021), we showed that the method is quantitatively accurate in reproducing experimental data for all the interphase chromosomes.

Having demonstrated this consistency, we used computationally simple data-driven predictions to analyze HCT-116 and mouse liver cell lines for which Hi-C data with and without cohesin rather than perform multiple laborious polymer simulations.

Please see below for our response to specific comments.1. It is confusing that the authors change continuously their label for describing B-A and A-B switches. They should choose one expression. I think that the label "switch" between A and B is more precise than "mismatch".

We have revised the text to make it consistent. Now it all reads “A-B”. Yes, the suggestion that we use switch is good but we think that mismatch is more concise. We trust that this Reviewer will indulge us on this point.

2. In the Abstract, the authors mention HCT-116 cells but do not specify which cells are these.

We have changed “HCT-116” in the abstract to “human colorectal carcinoma cell line”.

3. In the Abstract, it is unclear what the authors mean by "without any parameters"

In the theoretically based HIPPS method, there is no “free” parameter. In other words, the only parameter is uniquely determined. To avoid confusion, we have removed “without any parameters” from abstract.

4. In Results, what do the authors mean by 16% (26%)?

This refers the percentage of how many TADs are preserved after Nipbl and RAD21 removal in mouse and HCT-116 cells, respectively. Using TopDom method, we identified TAD boundaries in Wild and cohesin-depleted cells. There are 16% (959 out of 4176 – Fig. 1a) and 26% (1266 out of 4733 – Fig. 1b) of TADs are preserved after Nipbl and RAD21 removal in mouse and HCT-116 cells, respectively. We removed the percentages in the revised version.

5. In Results, the authors mention "more importantly, we did tune the value of any parameter to fit the experimental CMs". Did they mean that instead they didn't tune any parameter?

We apologize for the confusion. In the CCM, there is a single controlled parameter. We have changed the sentence to reflect this correctly.

6. In Results, section "CCM simulations reproduce wild-type Hi-C maps", Kullback-Leibler (KL) divergence is used to assess the correlation between two loci, but it is unclear what the value 0.04 stands for; is it a good or a bad correlation?

The value for Kullback-Leibler divergence can vary from 0 to infinity with 0 give the perfect correlation. Thus, 0.04 means that the correlation is excellent.

7. The authors use two techniques to obtain 3D structures, one is CCM, which takes the cohesin as constraints, and another is HIPPS, which reconstructs from Hi-C maps. Both seem to have good agreement with the Hi-C contact maps. However, did the authors compare the CCM with the HIPPS 3D structures?

This is detailed in response at the start of the reply to this Reviewer. As detailed in this response as well in the main text we used the CCM to generate hypotheses for the origin of P-TADs. In the process, we established the accuracy of CCM, which gives us confidence about the hypotheses. As explained above and emphasized in the revised version, CCM simulations are time consuming whereas generating 3D structures using HIPPS is computationally simple. Because HIPPS is also accurate, we used it to analyze the Hi-C data on mouse liver, HCT-116 as well as Micro-Data on mESC.

In our paper in Phys Rev X 11: 011051 (2021) we showed that HIPPS reproduces Hi-C data. In the current manuscript, we showed in Appendix Fig. 2 and Fig. 3 as well as in a study in 2018 (Shi and Thirumalai, Nat Comm.) that CCM is accurate as well. Thus, there is little doubt about the accuracies of the methods that we have developed.

8. In Results, section "P-TADs have prominent spatial domain boundaries", the authors constructed individual spatial distance matrices (DMs) using 10,000 simulated 3D structures. What are the differences among these 10,000 simulations? Do they start them with different initial structures?

The structures are generated using HIPPS which is data-driven method that uses Hi-C contact map as constraints. The method, which uses the maximum entropy theory, samples from a distribution that describe the structural ensemble of chromosome. The 10,000 structures are randomly sampled and are independent from each other. The HIPPS method is not a simulation, and hence the issue of initial structures does not arise.

9. In Methods, when the authors mention the "unknown parameter", do they use one parameter for all simulations (+/- cohesin) or is this parameter different for each system? Would this change the results?

We apologize for the confusion. The “unknown parameter” is the energy scale ε that describes the interaction strength between chromosome loci. We have revised the text in the method (page 27) to clarify it. The same value of ε is used for all CCM simulation with or without cohesin.

10. In Methods, when the authors perform DBSCAN clustering, they mention that they optimize the clustering parameters for each system. However, if they want to compare between different systems, the clustering parameters should be the same.

The purpose of DBSCAN is to capture the spatial clustering topology of chromosome loci. However, different cell types and chromosomes may have different overall density, which will impact the average distance between loci. If using the same parameters, such global changes will impact the result of clustering most and the intended spatial clustering topology can be distorted. Hence, we tune the clustering parameter for each system in order to ignore the global effect but only capture the local and topology of clustering of chromosome loci.

Grammar comments:1. "structures, with sharp boundaries are present, at.."

We thank the reviewer for pointing out the error. We have fixed it.

2. "Three headlines emerge from these studies are:"

We have fixed it.

3. "both the cell lines"

We have fixed it.